# Quasi-Newton Methods for Saddle Point Problems

**Chengchang Liu**
Department of Computer Science & Engineering
The Chinese University of Hong Kong
7liuchengchang@gmail.com

**Luo Luo**[*]
School of Data Science
Fudan University
luoluo@fudan.edu.cn

## Abstract

This paper studies quasi-Newton methods for strongly-convex-strongly-concave saddle point problems. We propose random Broyden family updates, which have explicit local superlinear convergence rate of $\mathcal{O}((1 - 1/(d\varkappa^2))^{k(k-1)/2})$, where $d$ is the dimension of the problem, $\varkappa$ is the condition number and $k$ is the number of iterations. The design and analysis of proposed algorithm are based on estimating the square of indefinite Hessian matrix, which is different from classical quasi-Newton methods in convex optimization. We also present two specific Broyden family algorithms with BFGS-type and SR1-type updates, which enjoy the faster local convergence rate of $\mathcal{O}((1 - 1/d)^{k(k-1)/2})$. Our numerical experiments show proposed algorithms outperform classical first-order methods.

## 1 Introduction

In this paper, we focus on the following smooth saddle point problem

$$\min_{\mathbf{x} \in \mathbb{R}^{d_x}} \max_{\mathbf{y} \in \mathbb{R}^{d_y}} f(\mathbf{x}, \mathbf{y}), \tag{1}$$

where $f$ is strongly-convex in $\mathbf{x}$ and strongly-concave in $\mathbf{y}$. We target to find the saddle point $(\mathbf{x}^*, \mathbf{y}^*)$ which holds that

$$f(\mathbf{x}^*, \mathbf{y}) \leq f(\mathbf{x}^*, \mathbf{y}^*) \leq f(\mathbf{x}, \mathbf{y}^*)$$

for all $\mathbf{x} \in \mathbb{R}^{d_x}$ and $\mathbf{y} \in \mathbb{R}^{d_y}$. This formulation is widely used in game theory [1, 39], AUC maximization [15, 43], fairness-aware machine learning [44], robust optimization [2, 11, 13, 36], empirical risk minimization [46] and reinforcement learning [10].

There are a great number of first-order optimization algorithms for solving problem (1), including extragradient method [18, 38], optimistic gradient descent ascent [7, 28], proximal point method [31] and dual extrapolation [24]. These algorithms iterate with first-order oracle and achieve linear convergence. Lin et al. [21], Wang and Li [40] used Catalyst acceleration to reduce the complexity for unbalanced saddle point problem, nearly matching the lower bound of first-order algorithms [26, 45] under some specific assumptions. Compared with first-order methods, second-order methods usually enjoy superior convergence in numerical optimization. Huang et al. [16] extended cubic regularized Newton (CRN) method [23, 24] to solve saddle point problem (1), which has quadratic local convergence. However, each iteration of CRN requires accessing the exact Hessian matrix and solving the corresponding linear systems. These steps arise $\mathcal{O}(d_x^3 + d_y^3)$ time complexity, which is too expensive for high dimensional problems.

Quasi-Newton methods [3–5, 8, 37] are popular ways to avoid accessing exact second-order information applied in standard Newton methods. They approximate the Hessian matrix based on the Broyden family updating formulas [3], which significantly reduces the computational cost. These algorithms

---

[*]The corresponding author

36th Conference on Neural Information Processing Systems (NeurIPS 2022).

Table 1: We summarize the convergence behaviors of proposed algorithms for solving saddle point problem in the view of gradient norm $\lambda_{k+k_0} \overset{\text{def}}{=} \|\nabla f(\mathbf{z}_{k+k_0})\|$ after $(k+k_0)$ iterations, where $d \overset{\text{def}}{=} d_x + d_y$ is dimensions of the problem and $\varkappa$ is the condition number. The results come from 3.18 and the upper bound holds with high probability at least $1 - \delta$.

| Algorithms | Upper Bound of $\lambda_{k+k_0}$ | $k_0$ |
|---|---|---|
| Broyden (Alg. 5) | $\left(1 - \frac{1}{d\varkappa^2+1}\right)^{k(k-1)/2} \left(\frac{1}{2}\right)^k \left(1 - \frac{1}{4\varkappa^2}\right)^{k_0}$ | $\mathcal{O}\left(d\varkappa^2 \ln\left(\frac{d\varkappa}{\delta}\right)\right)$ |
| BFGS/SR1 (Alg. 6 and 7) | $\left(1 - \frac{1}{d+1}\right)^{k(k-1)/2} \left(\frac{1}{2}\right)^k \left(1 - \frac{1}{4\varkappa^2}\right)^{k_0}$ | $\mathcal{O}\left((d + \varkappa^2) \ln\left(\frac{d\varkappa}{\delta}\right)\right)$ |

are well studied for convex optimization. The famous quasi-Newton methods including Davidon-Fletcher-Powell (DFP) method [8, 12], Broyden-Fletcher-Goldfarb-Shanno (BFGS) method [4, 5, 37] and symmetric rank 1 (SR1) method [3, 8] enjoy local superlinear convergence [6, 9, 29] when the objective function is strongly-convex. Recently, Rodomanov and Nesterov [32, 33, 34] proposed greedy and random variants of quasi-Newton methods, which first achieves non-asymptotic superlinear convergence. Later, Lin et al. [20] established a better convergence rate which is condition-number-free. Jin and Mokhtari [17], Ye et al. [42] showed the non-asymptotic superlinear convergence rate also holds for classical DFP, BFGS and SR1 methods.

In this paper, we study quasi-Newton methods for saddle point problem (1). Note that when the Hessian matrix of the objective function is indefinite, the existing Broyden family update formulas and their convergence analysis cannot be applied directly. To overcome this issue, we propose a variant framework of random quasi-Newton methods for saddle point problems, which approximates the square of the Hessian matrix during the iteration. Our theoretical analysis characterizes the convergence rate by the gradient norm, rather than the weighted norm of gradient used in convex optimization [17, 20, 32–34, 42]. We summarize the theoretical results for proposed algorithms in Table 1. The local convergence behaviors for all of the algorithms have two periods. The first period has $k_0$ iterations with a linear convergence rate $\mathcal{O}((1 - 1/\varkappa^2)^{k_0})$. The second one enjoys superlinear convergence, that is,

- For general Broyden family methods, we have $\mathcal{O}\left(\left(1 - 1/(d\varkappa^2 + 1)\right)^{k(k-1)/2}\right)$.

- For BFGS method and SR1 method, we have $\mathcal{O}\left(\left(1 - 1/(d + 1)\right)^{k(k-1)/2}\right)$, which is condition-number-free.

**Paper Organization** In Section 2, we state the notation and preliminaries of this paper. In Section 3, we first propose random quasi-Newton methods for quadratic saddle point problem which enjoys local superlinear convergence. Then we extend it to solve general strongly-convex-strongly-concave saddle point problems. In Section 5, we provide numerical experiments to validate our algorithms on popular machine learning models. All proofs, experiment details and extensions are deferred to appendix.

## 2 Notation and Preliminaries

We use $\|\cdot\|$ to present spectral norm and Euclidean norm of matrix and vector respectively. We denote the standard basis for $\mathbb{R}^d$ by $\{\mathbf{e}_1, \ldots, \mathbf{e}_d\}$ and let $\mathbf{I}$ be the identity matrix. The trace of a square matrix is denoted by $\mathrm{tr}(\cdot)$. Given two positive definite matrices $\mathbf{G}$ and $\mathbf{H}$, we define their inner product as $\langle \mathbf{G}, \mathbf{H} \rangle \overset{\text{def}}{=} \mathrm{tr}(\mathbf{GH})$.

We introduce the quantity $\sigma_{\mathbf{H}}(\mathbf{G}) \overset{\text{def}}{=} \langle \mathbf{H}^{-1}, \mathbf{G} - \mathbf{H} \rangle = \langle \mathbf{H}^{-1}, \mathbf{G} \rangle - d$, which is used to measure how well does matrix $\mathbf{G}$ approximate matrix $\mathbf{H}$. If we further suppose $\mathbf{G} \succeq \mathbf{H}$, then according to Rodomanov and Nesterov [32] it holds that $\mathbf{G} - \mathbf{H} \preceq \langle \mathbf{H}^{-1}, \mathbf{G} - \mathbf{H} \rangle \mathbf{H} = \sigma_{\mathbf{H}}(\mathbf{G})\mathbf{H}$.

Using the notation of problem (1), we let $\mathbf{z} = [\mathbf{x}; \mathbf{y}] \in \mathbb{R}^d$ where $d \overset{\text{def}}{=} d_x + d_y$ and denote the gradient and Hessian matrix of $f$ at $(\mathbf{x}, \mathbf{y})$ as $\mathbf{g}(\mathbf{z}) \in \mathbb{R}^d$ and $\hat{\mathbf{H}}(\mathbf{z}) \in \mathbb{R}^{d \times d}$.

We suppose the saddle point problem (1) satisfies the following assumptions.

**Assumption 2.1.** The objective function $f(\mathbf{x}, \mathbf{y})$ is twice differentiable and has $L$-Lipschitz continuous gradient and $L_2$-Lipschitz continuous Hessian, i.e., there exists constants $L > 0$ and $L_2 > 0$ such that for any $\mathbf{z} = [\mathbf{x}; \mathbf{y}], \mathbf{z}' = [\mathbf{x}'; \mathbf{y}'] \in \mathbb{R}^d$, we have $\|\mathbf{g}(\mathbf{z}) - \mathbf{g}(\mathbf{z}')\| \leq L\|\mathbf{z} - \mathbf{z}'\|$ and $\|\hat{\mathbf{H}}(\mathbf{z}) - \hat{\mathbf{H}}(\mathbf{z}')\| \leq L_2\|\mathbf{z} - \mathbf{z}'\|$.

**Assumption 2.2.** The objective function $f(\mathbf{x}, \mathbf{y})$ is twice differentiable, $\mu$-strongly-convex in $\mathbf{x}$ and $\mu$-strongly-concave in $\mathbf{y}$, i.e., there exists constant $\mu > 0$ such that $\nabla^2_{\mathbf{xx}} f(\mathbf{x}, \mathbf{y}) \succeq \mu\mathbf{I}$ and $\nabla^2_{\mathbf{yy}} f(\mathbf{x}, \mathbf{y}) \preceq -\mu\mathbf{I}$ for any $(\mathbf{x}, \mathbf{y}) \in \mathbb{R}^{d_x} \times \mathbb{R}^{d_y}$.

The $L$-Lipschitz continuous of gradient means the spectral norm of Hessian matrix $\hat{\mathbf{H}}(\mathbf{z})$ can be upper bounded, that is $\|\hat{\mathbf{H}}(\mathbf{z})\| \leq L$. Additionally, the condition number of the objective function is defined as $\varkappa \overset{\text{def}}{=} L/\mu$.

# 3 Quasi-Newton Methods for Saddle Point Problems

In this section, we focus on designing quasi-Newton methods for saddle point problem and showing their superlinear local convergence rate. The update rule of standard Newton's method can be written as $\mathbf{z}_+ = \mathbf{z} - (\hat{\mathbf{H}}(\mathbf{z}))^{-1}\mathbf{g}(\mathbf{z})$ for solving problem (1). It has quadratic local convergence, but takes $\mathcal{O}(d^3)$ time complexity per iteration. For convex minimization, quasi-Newton methods including BFGS/SR1 [3–5, 8, 37] and their variants [20, 32, 33, 42] focus on approximating the Hessian which can reduce the computational cost to $\mathcal{O}(d^2)$ for each round. However, all these algorithms and related convergence analysis are based on the assumption that the Hessian matrix is positive definite, which is not suitable for our saddle point problems since $\hat{\mathbf{H}}(\mathbf{z})$ is indefinite.

We introduce the auxiliary matrix $\mathbf{H}(\mathbf{z})$ be the square of Hessian $\mathbf{H}(\mathbf{z}) \overset{\text{def}}{=} (\hat{\mathbf{H}}(\mathbf{z}))^2$. The following lemma shows that $\mathbf{H}(\mathbf{z})$ is positive definite.

**Lemma 3.1.** *Under Assumption 2.1 and 2.2, we have $\mu^2\mathbf{I} \preceq \mathbf{H}(\mathbf{z}) \preceq L^2\mathbf{I}$ for all $\mathbf{z} \in \mathbb{R}^d$.*

Hence, we can reformulate the update of Newton's method by

$$
\begin{aligned}
\mathbf{z}_+ &= \mathbf{z} - [(\hat{\mathbf{H}}(\mathbf{z}))^2]^{-1}\hat{\mathbf{H}}(\mathbf{z})\mathbf{g}(\mathbf{z}) \\
&= \mathbf{z} - \mathbf{H}(\mathbf{z})^{-1}\hat{\mathbf{H}}(\mathbf{z})\mathbf{g}(\mathbf{z}).
\end{aligned}
\tag{2}
$$

Then it is natural to characterize the second-order information by estimating the auxiliary matrix $\mathbf{H}(\mathbf{z})$, rather than the indefinite Hessian $\hat{\mathbf{H}}(\mathbf{z})$. If one can obtain a symmetric positive definite matrix $\mathbf{G} \in \mathbb{R}^{d \times d}$ as an estimator for $\mathbf{H}(\mathbf{z})$, the update rule of (2) can be approximated by

$$
\mathbf{z}_+ = \mathbf{z} - \mathbf{G}^{-1}\hat{\mathbf{H}}(\mathbf{z})\mathbf{g}(\mathbf{z}).
\tag{3}
$$

The remainder of this section introduce strategies to construct $\mathbf{G}$, resulting the quasi-Newton methods for saddle point problem with local superlinear convergence. We should point out the implementation of iteration (3) is unnecessary to construct Hessian matrix $\hat{\mathbf{H}}(\mathbf{z})$ explicitly, since we are only interested in the Hessian-vector product $\hat{\mathbf{H}}(\mathbf{z})\mathbf{g}(\mathbf{z})$ which can be computed efficiently [27, 35].

## 3.1 The Broyden Family Updates

We first review some basic results for quasi-Newton methods in convex optimization. We introduce the Broyden family [25, Section 6.3] of quasi-Newton updates for approximating an positive definite matrix $\mathbf{H} \in \mathbb{R}^{d \times d}$ by using the information of current estimator $\mathbf{G} \in \mathbb{R}^{d \times d}$.

**Definition 3.2.** Suppose two positive definite matrices $\mathbf{H} \in \mathbb{R}^{d \times d}$ and $\mathbf{G} \in \mathbb{R}^{d \times d}$ satisfy $\mathbf{H} \preceq \mathbf{G}$. For any $\mathbf{u} \in \mathbb{R}^d$, if $\mathbf{Gu} = \mathbf{Hu}$, we define $\mathrm{Broyd}_\tau (\mathbf{G}, \mathbf{H}, \mathbf{u}) \overset{\mathrm{def}}{=} \mathbf{H}$. Otherwise, we define

$$
\begin{aligned}
\mathrm{Broyd}_\tau (\mathbf{G}, \mathbf{H}, \mathbf{u}) \overset{\mathrm{def}}{=} &(1 - \tau) \left[ \mathbf{G} - \frac{(\mathbf{G} - \mathbf{H})\mathbf{u}\mathbf{u}^\top (\mathbf{G} - \mathbf{H})}{\mathbf{u}^\top (\mathbf{G} - \mathbf{H})\mathbf{u}} \right] \\
&+ \tau \left[ \mathbf{G} - \frac{\mathbf{Hu}\mathbf{u}^\top \mathbf{G} + \mathbf{Gu}\mathbf{u}^\top \mathbf{H}}{\mathbf{u}^\top \mathbf{Hu}} + \left( \frac{\mathbf{u}^\top \mathbf{Gu}}{\mathbf{u}^\top \mathbf{Hu}} + 1 \right) \frac{\mathbf{Hu}\mathbf{u}^\top \mathbf{H}}{\mathbf{u}^\top \mathbf{Hu}} \right].
\end{aligned}
\tag{4}
$$

The different choices of parameter $\tau$ for above formula contain several popular quasi-Newton updates:

- For $\tau = \mathbf{u}^\top \mathbf{Hu} / \left( \mathbf{u}^\top \mathbf{Gu} \right) \in [0, 1]$, it corresponds to the BFGS update

$$
\mathrm{BFGS}(\mathbf{G}, \mathbf{H}, \mathbf{u}) \overset{\mathrm{def}}{=} \mathbf{G} - \frac{\mathbf{Gu}\mathbf{u}^\top \mathbf{G}}{\mathbf{u}^\top \mathbf{Gu}} + \frac{\mathbf{Hu}\mathbf{u}^\top \mathbf{H}}{\mathbf{u}^\top \mathbf{Hu}}.
\tag{5}
$$

- For $\tau = 0$, it corresponds to the SR1 update

$$
\mathrm{SR1}(\mathbf{G}, \mathbf{H}, \mathbf{u}) \overset{\mathrm{def}}{=} \mathbf{G} - \frac{(\mathbf{G} - \mathbf{H})\mathbf{u}\mathbf{u}^\top (\mathbf{G} - \mathbf{H})}{\mathbf{u}^\top (\mathbf{G} - \mathbf{H})\mathbf{u}}.
\tag{6}
$$

Now we introduce the update rule [20, 32] by choosing $\mathbf{u}$ as

$$
\mathbf{u} \sim (\mathbf{0}, \mathbf{I}) \qquad \text{or} \qquad \mathbf{u} \sim \mathrm{Unif}\left( \mathcal{S}^{d-1} \right).
\tag{7}
$$

The following lemma shows applying the update rule (4) with (7) leads to a new estimator with tighter error bound in the measure of $\sigma_\mathbf{H}(\cdot)$.

**Lemma 3.3** (Modified from Lin et al. [20, Theorem 3.1]). *Suppose two positive definite matrices* $\mu^2 \mathbf{I} \preceq \mathbf{H} \preceq L^2 \mathbf{I}$ *and* $\mathbf{G} \in \mathbb{R}^{d \times d}$ *satisfy that* $\mathbf{H} \preceq \mathbf{G}$. *Let* $\mathbf{G}_+ = \mathrm{Broyd}_\tau (\mathbf{G}, \mathbf{H}, \mathbf{u})$, *where* $\mathbf{u}$ *is chosen random method as* (7), *then for any* $\tau \in [0, 1]$, *we have* $\mathbb{E}\left[ \sigma_\mathbf{H}(\mathbf{G}_+) \right] \leq \left( 1 - 1/(d\varkappa^2) \right) \sigma_\mathbf{H}(\mathbf{G})$

For specific Broyden family updates BFGS and SR1 shown in (5) and (6), the update can achieve a better convergence result. Concretely, for BFGS method, we first find $\mathbf{L}$ such that $\mathbf{G}^{-1} = \mathbf{L}^\top \mathbf{L}$, where $\mathbf{L}$ is an upper triangular matrix. This step can be implemented with $\mathcal{O}(d^2)$ complexity [20, Proposition 1]. We present the subroutine for factorizing $\mathbf{G}^{-1}$ in Algorithm 1 and give its detailed implementation in the appendix. And we use the direction $\mathbf{L}^\top \mathbf{u}$ instead of $\mathbf{u}$ for the BFGS update. Applying the BFGS update rule (5) with formula (7), we obtain a condition-number-free result as follows.

**Lemma 3.4** (Modified from Lin et al. [20, Theorem 4.2]). *Suppose two positive definite matrices* $\mathbf{H} \in \mathbb{R}^{d \times d}$ *and* $\mathbf{G} \in \mathbb{R}^{d \times d}$ *satisfy* $\mathbf{H} \preceq \mathbf{G}$. *Let* $\mathbf{G}_+ = \mathrm{BFGS}\left( \mathbf{G}, \mathbf{H}, \mathbf{L}^\top \mathbf{u} \right)$, *where* $\mathbf{u}$ *is chosen by the random method in* (7) *and* $\mathbf{L}$ *is an upper triangular matrix such that* $\mathbf{G}^{-1} = \mathbf{L}^\top \mathbf{L}$. *Then we have* $\mathbb{E}\left[ \sigma_\mathbf{H}(\mathbf{G}_+) \right] \leq (1 - 1/d)\, \sigma_\mathbf{H}(\mathbf{G})$.

*Remark* 3.5. Note that the step of conducting $\mathbf{G}^{-1} = \mathbf{L}^\top \mathbf{L}$ requires QR decomposition of rank-1 change matrix which requires $\mathcal{O}(26d^2)$ flops [14, Section 12.5.1]. We do not recommend using this BFGS update strategy in practice when $n$ is large.

The convergence of the SR1 update can be characterized by the measure $\tau_\mathbf{H}(\mathbf{G}) \overset{\mathrm{def}}{=} \mathrm{tr}(\mathbf{G} - \mathbf{H})$. Applying the SR1 update rule (6) with formula (7), the convergence also holds a condition-number-free result.

**Lemma 3.6** (Modified from Lin et al. [20, Theorem 4.1]). *Suppose two positive definite matrices* $\mathbf{H} \in \mathbb{R}^{d \times d}$ *and* $\mathbf{G} \in \mathbb{R}^{d \times d}$ *satisfy* $\mathbf{H} \preceq \mathbf{G}$. *Let* $\mathbf{G}_+ = \mathrm{SR1}\left( \mathbf{G}, \mathbf{H}, \mathbf{u} \right)$, *where* $\mathbf{u}$ *is chosen by the random method in* (7). *Then we have* $\mathbb{E}\left[ \tau_\mathbf{H}\left( \mathbf{G}_+ \right) \right] \leq (1 - 1/d)\, \tau_\mathbf{H}(\mathbf{G})$.

## 3.2 Algorithms for Quadratic Saddle Point Problems

We now solve simple quadratic saddle point problem that $f(\mathbf{x}, \mathbf{y}) = \frac{1}{2}\mathbf{z}^\top \mathbf{Az} - \mathbf{b}^\top \mathbf{z}$ in (1) where $\mathbf{z} = [\mathbf{x}; \mathbf{y}]$, $\mathbf{b} \in \mathbb{R}^d$, $\mathbf{A} \in \mathbb{R}^{d \times d}$ is symmetric and $d = d_x + d_y$. We suppose $\mathbf{A}$ could be partitioned as $\mathbf{A} = \begin{bmatrix} \mathbf{A_{xx}} & \mathbf{A_{xy}} \\ \mathbf{A_{yx}} & \mathbf{A_{yy}} \end{bmatrix}$ where the sub-matrices $\mathbf{A_{xx}} \in \mathbb{R}^{d_x \times d_x}$, $\mathbf{A_{xy}} \in \mathbb{R}^{d_x \times d_y}$, $\mathbf{A_{yx}} \in \mathbb{R}^{d_y \times d_x}$

---

**Algorithm 1** Fast-Chol($\mathbf{H}, \mathbf{L}, \mathbf{u}$)

---

1: **Input:** positive definite matrix $\mathbf{H} \in \mathbb{R}^{d \times d}$, upper triangular matrix $\mathbf{L} \in \mathbb{R}^d$, direction $\mathbf{u} \in \mathbb{R}^d$

2: Using QR-decomposition to obtain $[\mathbf{Q}, \mathbf{R}] = \mathrm{QR}\left(\mathbf{L}\left(\mathbf{I} - \mathbf{H}\mathbf{u}\mathbf{u}^\top/(\mathbf{u}^\top\mathbf{H}\mathbf{u})\right)\right)$

3: Calculate $\mathbf{v} = \mathbf{u}/\sqrt{\mathbf{u}^\top\mathbf{H}\mathbf{u}}$ and $[\mathbf{Q}', \mathbf{R}'] = \mathrm{QR}(\begin{bmatrix} \mathbf{v} & \mathbf{R}^\top \end{bmatrix}^\top)$

4: **Output:** $\hat{\mathbf{L}} = \mathbf{R}'$

---

---

**Algorithm 2** Random-Broyden-Quadratic

---

1: **Input:** $\mathbf{z}_0 \in \mathbb{R}^d$, $\mathbf{G}_0 = L^2\mathbf{I}$ and $\tau_k \in [0, 1]$

2: **for** $k = 0, 1, \ldots$

3: $\quad \mathbf{z}_{k+1} = \mathbf{z}_k - \mathbf{G}_k^{-1}\hat{\mathbf{H}}\mathbf{g}(\mathbf{z}_k)$

4: $\quad$ Randomly choose $\mathbf{u}_k$ from (7) and update $\mathbf{G}_{k+1} = \mathrm{Broyd}_{\tau_k}\left(\mathbf{G}_k, \mathbf{H}, \mathbf{u}_k\right)$

5: **end for**

---

and $\mathbf{A_{yy}} \in \mathbb{R}^{d_y \times d_y}$ satisfy $\mathbf{A_{xx}} \succeq \mu\mathbf{I}$, $\mathbf{A_{yy}} \preceq -\mu\mathbf{I}$ and $\|\mathbf{A}\| \leq L$. Using notations introduced in Section 2, we have $\mathbf{z} = [\mathbf{x}; \mathbf{y}]$, $\mathbf{g}(\mathbf{z}) = \mathbf{A}\mathbf{z} - \mathbf{b}$, $\hat{\mathbf{H}} \stackrel{\text{def}}{=} \hat{\mathbf{H}}(\mathbf{z}) = \mathbf{A}$ and $\mathbf{H} \stackrel{\text{def}}{=} \mathbf{H}(\mathbf{z}) = \mathbf{A}^2$.

We present the detailed procedure of random quasi-Newton methods for quadratic saddle point problem by using the Broyden family update, BFGS update and SR1 update in Algorithm 2, 3 and 4 respectively.

We define $\lambda_k$ as the gradient norm at $\mathbf{z}_k$ for our convergence analysis, that is $\lambda_k \stackrel{\text{def}}{=} \|\mathbf{g}(\mathbf{z}_k)\|$. The definition of $\lambda_k$ in this paper is different from the measure used in convex optimization [20, 32][2], but it also holds the similar property as follows.

**Lemma 3.7.** *Assume we have $\eta_k \geq 1$ and $\mathbf{G}_k \in \mathbb{R}^{d \times d}$ such that $\mathbf{H} \preceq \mathbf{G}_k \preceq \eta_k\mathbf{H}$ for Algorithm 2-4, then we have $\lambda_{k+1} \leq (1 - 1/\eta_k)\lambda_k$.*

The next theorem states the assumptions of Lemma 3.7 always holds with $\eta_k = \varkappa^2 \geq 1$, which means $\lambda_k$ converges to 0 linearly.

**Theorem 3.8.** *For all $k \geq 0$, Algorithm 2, 3, 4 hold that $\lambda_k \leq \left(1 - \frac{1}{\varkappa^2}\right)^k \lambda_0$ and $\mathbf{H} \preceq \mathbf{G}_k \preceq \varkappa$.*

Lemma 3.7 also implies superlinear convergence can be obtained if there exists $\eta_k$ which converges to 1. Applying Lemma 3.3, 3.4 and 3.6, we can show it holds for proposed algorithms.

**Theorem 3.9.** *Solving quadratic saddle point problem by the proposed quasi-Newton Algorithms, for all $k \geq 0$, we have:*

1. *For the Broyden family method (Algorithm 2), we have $\mathbb{E}\left[\lambda_{k+1}/\lambda_k\right] \leq \left(1 - 1/(d\varkappa^2)\right)^k d\varkappa^2$.*

2. *For the BFGS method (Algorithm 3), we have $\mathbb{E}\left[\lambda_{k+1}/\lambda_k\right] \leq (1 - 1/d)^k d\varkappa^2$.*

3. *For the SR1 method (Algorithm 4), we have $\mathbb{E}\left[\lambda_{k+1}/\lambda_k\right] \leq (1 - k/d) d\varkappa^4$.*

Combining the results of Theorem 3.8 and 3.9, we achieve the two-stages convergence behavior, that is, the algorithm has the global linear convergence and local superlinear convergence. We leave the formal description in appendix.

### 3.3 Algorithms for General Saddle Point Problems

In this section, we consider the general saddle point problem where $f(\mathbf{x}, \mathbf{y})$ in (1) satisfies Assumption 2.1 and 2.2. We propose quasi-Newton methods for solving the problem with local superlinear convergence and $\mathcal{O}(d^2)$ time complexity for each iteration.

---

[2]In later section, we will see the measure $\lambda_k \stackrel{\text{def}}{=} \|\mathbf{g}_k\|$ is suitable to convergence analysis of quasi-Newton methods for saddle point problems.

| **Algorithm 3** Random-BFGS-Quadratic | **Algorithm 4** Random-SR1-Quadratic |
|---|---|
| 1: **Input:** $\mathbf{z}_0 \in \mathbb{R}^d$, $\mathbf{G}_0 = L^2\mathbf{I}$, $\mathbf{L}_0 = L^{-1}\mathbf{I}$ | 1: **Input:** $\mathbf{z}_0 \in \mathbb{R}^d$, $\mathbf{G}_0 = L^2\mathbf{I}$ |
| 2: **for** $k = 0, 1 \ldots$ |     $K \leq d + 1$ |
| 3:    $\mathbf{z}_{k+1} = \mathbf{z}_k - \mathbf{G}_k^{-1}\hat{\mathbf{H}}\mathbf{g}(\mathbf{z}_k)$ | 2: **for** $k = 0, 1 \ldots, K$ |
| 4:    Randomly choose $\tilde{\mathbf{u}}_k$ from (7) | 3:    $\mathbf{z}_{k+1} = \mathbf{z}_k - \mathbf{G}_k^{-1}\hat{\mathbf{H}}\mathbf{g}(\mathbf{z}_k)$ |
| 5:    $\mathbf{u}_k = \mathbf{L}_k^\top \tilde{\mathbf{u}}_k$ | 4:    Randomly choose $\mathbf{u}_k$ from (7) |
| 6:    $\mathbf{G}_{k+1} = \mathrm{BFGS}\left(\mathbf{G}_k, \mathbf{H}, \mathbf{u}_k\right)$ | 5:    $\mathbf{G}_{k+1} = \mathrm{SR1}\left(\mathbf{G}_k, \mathbf{H}, \mathbf{u}_k\right)$ |
| 7:    $\mathbf{L}_{k+1} = \mathrm{Fast\text{-}Chol}(\mathbf{H}, \mathbf{L}_k, \mathbf{u}_k)$ | 6: **end for** |
| 8: **end for** | |

### 3.3.1 Algorithms

The key idea of designing quasi-Newton methods for saddle point problems is approximating the auxiliary matrix $\mathbf{H}(\mathbf{z}) \stackrel{\text{def}}{=} \left(\hat{\mathbf{H}}(\mathbf{z})\right)^2$ to characterize the second-order information. Since the Hessian of $f$ is Lipschitz continuous and bounded by Assumption 2.1 and 2.2, which means the auxiliary matrix operator $\mathbf{H}(\mathbf{z})$ is also Lipschitz continuous.

**Lemma 3.10.** *Under Assumption 2.1 and 2.2, we have $\mathbf{H}(\mathbf{z})$ is $2LL_2$-Lipschitz continuous.*

Combining Lemma 3.1 and 3.10, we achieve the following properties of $\mathbf{H}(\mathbf{z})$, which analogize the strongly self-concordance in convex optimization [32].

**Lemma 3.11.** *Under Assumption 2.1 and 2.2, for all $\mathbf{z}, \mathbf{z}', \mathbf{w} \in \mathbb{R}^d$, the auxiliary matrix operator $\mathbf{H}(\cdot)$ satisfies $\mathbf{H}(\mathbf{z}) - \mathbf{H}(\mathbf{z}') \preceq M\|\mathbf{z} - \mathbf{z}'\|\mathbf{H}(\mathbf{w})$, where $M = 2\varkappa^2 L_2/L$.*

**Corollary 3.12.** *Let $\mathbf{z}, \mathbf{z}' \in \mathbb{R}^d$ and $r = \|\mathbf{z}' - \mathbf{z}\|$. Suppose the objective function $f$ satisfies Assumption 2.1 and 2.2, then for all $\mathbf{z}, \mathbf{z}' \in \mathbb{R}^d$ and $M = 2\varkappa^2 L_2/L$, the auxiliary matrix operator $\mathbf{H}(\cdot)$ holds that*

$$\frac{\mathbf{H}(\mathbf{z})}{(1 + Mr)} \preceq \mathbf{H}(\mathbf{z}') \preceq (1 + Mr)\mathbf{H}(\mathbf{z})$$

Different from the quadratic case, the auxiliary matrix $\mathbf{H}(\mathbf{z})$ is not fixed for general saddle point problem. Based on the smoothness of $\mathbf{H}(\mathbf{z})$, we apply Corollary 3.12 to generalize Lemma 3.13 as follows.

**Lemma 3.13.** *Let $\mathbf{z} \in \mathbb{R}^d$ and $\mathbf{G} \in \mathbb{R}^{d \times d}$ be a positive definite matrix such that $\mathbf{H}(\mathbf{z}) \preceq \mathbf{G} \preceq \eta\mathbf{H}(\mathbf{z})$ for some $\eta \geq 1$. In addition, define $\mathbf{z}_+ \in \mathbb{R}^d$ and $r = \|\mathbf{z}_+ - \mathbf{z}\|$, then for all $\mathbf{u} \in \mathbb{R}^d$, $\tau \in [0, 1]$ and $M = 2\varkappa^2 L_2/L$, we have*

$$\mathbf{H}(\mathbf{z}_+) \preceq \mathrm{Broyd}_\tau(\tilde{\mathbf{G}}, \mathbf{H}(\mathbf{z}_+), \mathbf{u}) \preceq (1 + Mr)^2\eta\mathbf{H}(\mathbf{z}_+),$$

*where $\tilde{\mathbf{G}} \stackrel{\text{def}}{=} (1 + Mr)\mathbf{G} \succeq \mathbf{H}(\mathbf{z}_+)$.*

Lemma 3.13 implies it is reasonable to have the algorithms using $\mathbf{G}_{k+1} = \mathrm{Broyd}_{\tau_k}(\tilde{\mathbf{G}}_k, \mathbf{H}_{k+1}, \mathbf{u}_k)$ with $\tilde{\mathbf{G}}_k = (1 + Mr_k)\mathbf{G}_k$ and $r_k = \|\mathbf{z}_{k+1} - \mathbf{z}_k\|$. Similarly, we can also achieve $\mathbf{G}_{k+1}$ by such $\tilde{\mathbf{G}}_k$ for specific BFGS and SR1 update. Combining this with iteration (3), we propose several quasi-Newton methods for general strongly-convex-strongly-concave saddle point problems. The details are shown in Algorithm 5, 6 and 7 for Broyden family, BFGS and SR1 updates respectively.

### 3.3.2 Convergence Analysis

Let us consider the convergence guarantee for algorithms proposed in Section 3.3.1. We introduce the following notations to simplify the presentation. We let $\{\mathbf{z}_k\}$ be the sequence generated from Algorithm 5, 6 or 7 and denote

$$\mathbf{g}_k \stackrel{\text{def}}{=} \mathbf{g}(\mathbf{z}_k), \quad r_k \stackrel{\text{def}}{=} \|\mathbf{z}_{k+1} - \mathbf{z}_k\|, \quad \hat{\mathbf{H}}_k \stackrel{\text{def}}{=} \hat{\mathbf{H}}(\mathbf{z}_k) \quad \text{and} \quad \mathbf{H}_k \stackrel{\text{def}}{=} \left(\hat{\mathbf{H}}(\mathbf{z}_k)\right)^2.$$

We still use gradient norm $\lambda_k \stackrel{\text{def}}{=} \|\nabla f(\mathbf{z}_k)\|$ for analysis and establish the relationship between $\lambda_k$ and $\lambda_{k+1}$, which is shown in Lemma 3.14.

---

**Algorithm 5** Random-Broyden-General

1: **Input:** $\mathbf{z}_0 \in \mathbb{R}^d$, $\mathbf{G}_0 \succeq \mathbf{H}$, $\tau_k \in [0,1]$ and $M \geq 0$.

2: **for** $k = 0, 1 \ldots$

3:     $\mathbf{z}_{k+1} = \mathbf{z}_k - \mathbf{G}_k^{-1} \hat{\mathbf{H}}_k \mathbf{g}_k$

4:     Compute $r_k = \|\mathbf{z}_{k+1} - \mathbf{z}_k\|$    and    $\tilde{\mathbf{G}}_k = (1 + Mr_k)\mathbf{G}_k$

5:     Randomly choose $\mathbf{u}_k$ from (7) and update $\mathbf{G}_{k+1} = \text{Broyd}_{\tau_k}(\tilde{\mathbf{G}}_k, \mathbf{H}_{k+1}, \mathbf{u}_k)$.

6: **end for**

---

**Algorithm 6** Random-BFGS-General

1: **Input:** $\mathbf{z}_0 \in \mathbb{R}^d$, $\mathbf{G}_0 \succeq \mathbf{H}$, $M \geq 0$.
2: $\mathbf{L}_0 = \mathbf{G}_0^{-1/2}$
3: **for** $k = 0, 1 \ldots$
4:     $\mathbf{z}_{k+1} = \mathbf{z}_k - \mathbf{G}_k^{-1} \hat{\mathbf{H}}_k \mathbf{g}_k$
5:     $r_k = \|\mathbf{z}_{k+1} - \mathbf{z}_k\|$
6:     $\tilde{\mathbf{G}}_k = (1 + Mr_k)\mathbf{G}_k$
7:     $\tilde{\mathbf{L}}_k = \mathbf{L}_k / \sqrt{1 + Mr_k}$
8:     Randomly choose $\mathbf{u}_k$ from (7)
9:     $\mathbf{G}_{k+1} = \text{BFGS}(\tilde{\mathbf{G}}_k, \mathbf{H}_{k+1}, \tilde{\mathbf{L}}_k \mathbf{u}_k)$
10:    $\mathbf{L}_{k+1} = \text{Fast-Chol}(\mathbf{H}_{k+1}, \tilde{\mathbf{L}}_k, \tilde{\mathbf{L}}_k \mathbf{u}_k)$
11: **end for**

**Algorithm 7** Random-SR1-General

1: **Input:** $\mathbf{z}_0 \in \mathbb{R}^d$, $\mathbf{G}_0 \succeq \mathbf{H}$ and $M \geq 0$
2: **for** $k = 0, 1 \ldots$
3:     $\mathbf{z}_{k+1} = \mathbf{z}_k - \mathbf{G}_k^{-1} \hat{\mathbf{H}}_k \mathbf{g}_k$
4:     $r_k = \|\mathbf{z}_{k+1} - \mathbf{z}_k\|$
5:     $\tilde{\mathbf{G}}_k = (1 + Mr_k)\mathbf{G}_k$
6:     Randomly choose $\mathbf{u}_k$ from (7)
7:     $\mathbf{G}_{k+1} = \text{SR1}(\tilde{\mathbf{G}}_k, \mathbf{H}_{k+1}, \mathbf{u}_k)$
8: **end for**

---

**Lemma 3.14.** *Using Algorithm 5, 6 and 7, suppose we have $\mathbf{H}_k \preceq \mathbf{G}_k \preceq \eta_k \mathbf{H}_k$, for some $\eta_k \geq 1$ and let $\beta = L_2/(2\mu^2)$, then we have*

$$\lambda_{k+1} \leq \left(1 - \frac{1}{\eta_k}\right)\lambda_k + \beta \lambda_k^2 \quad \text{and} \quad r_k \leq \frac{\lambda_k}{\mu}.$$

Rodomanov and Nesterov [32, Lemma 4.3] derive a result similar to Lemma 3.14 for minimizing the strongly-convex function $\hat{f}(\cdot)$ on the different measure $\lambda_{\hat{f}}(\cdot) \stackrel{\text{def}}{=} \langle \nabla \hat{f}(\cdot), \nabla^2 \hat{f}(\cdot)^{-1} \nabla \hat{f}(\cdot)\rangle$.[3] Note that our algorithms are based on the iteration rule $\mathbf{z}_{k+1} = \mathbf{z}_k - \mathbf{G}_k^{-1} \hat{\mathbf{H}}_k \mathbf{g}_k$. Compared with quasi-Newton methods for convex optimization, there exists an additional term $\hat{\mathbf{H}}_k$ between $\mathbf{G}_k^{-1}$ and $\mathbf{g}_k$, which leads to the fact that the convergence analysis based on $\lambda_{\hat{f}}(\mathbf{z}_k)$ is difficult. Fortunately, we find using gradient norm $\lambda_k$ directly makes the analysis achievable.

For further analysis, we also denote $\sigma_k \stackrel{\text{def}}{=} \sigma_{\mathbf{H}_k}(\mathbf{G}_k) = \langle \mathbf{H}_k^{-1}, \mathbf{G}_k\rangle - d$.

Then we establish the linear convergence for the first period of iterations, which can be viewed as the extension of Theorem 3.8. Note that the following result holds for Algorithm 5, 6 and 7; and it does not depend on the choice of $\mathbf{u}_k$.

**Theorem 3.15.** *Using Algorithm 5, 6 and 7 by $\mathbf{G}_0 = L^2 \mathbf{I}$ and $M = 2\varkappa^2 L_2/L$, suppose the initial point $\mathbf{z}_0$ is sufficiently close to $\mathbf{z}^*$ such that $M\lambda_0/\mu \leq \ln b/(4b\varkappa^2)$ with $1 < b < 5$, then we have $\lambda_k \leq \left(1 - 1/2b\varkappa^2\right)^k \lambda_0$ and $\mathbf{H}_k \preceq \mathbf{G}_k \preceq \exp\left(2\sum_{i=0}^{k-1} \rho_i\right)\varkappa^2 \mathbf{H}_k \preceq b\varkappa^2 \mathbf{H}_k$ for all $k \geq 0$, where $\rho_k \stackrel{\text{def}}{=} M\lambda_k/\mu$.*

We now analyze how $\sigma_k$ changes after one iteration to show the local superlinear convergence for the Broyden family method (Algorithm 5) and the BFGS method (Algorithm 6). Recall that $\sigma_k$ is defined to measure how well does matrix $\mathbf{G}_k$ approximate $\mathbf{H}_k$.

**Lemma 3.16.** *Solving the general strongly-convex-strongly-concave saddle point problem* (1) *under Assumption 2.1 and 2.2 by our quasi-Newton algorithms and supposing the sequences $\mathbf{G}_0, \cdots, \mathbf{G}_k$ generated by the Algorithm 5 and 6 are given, then we have the following results for $\sigma_k$ for all $k$:*

---

[3]The original notations of Rodomanov and Nesterov [32] is minimizing the strongly-convex function $f(\cdot)$, to avoid ambiguity, we use notations $\hat{f}(\cdot)$ and $\lambda_{\hat{f}}$ to describe their work in this paper.

1. *The random Broyden family method (Algorithm 5) holds that*

$$\mathbb{E}\left[\sigma_{k+1}\right] \leq \left(1 - 1/(d\varkappa^2)\right)\left(1 + Mr_k\right)^2 \left(\sigma_k + 2dMr_k/(1 + Mr_k)\right).$$

2. *The random BFGS method (Algorithm 6) holds that*

$$\mathbb{E}\left[\sigma_{k+1}\right] \leq \left(1 - 1/d\right)\left(1 + Mr_k\right)^2 \left(\sigma_k + 2dMr_k/(1 + Mr_k)\right).$$

The analysis for the SR1 method is based on constructing $\eta_k$ such that $\eta_k = \mathrm{tr}(\mathbf{G}_k - \mathbf{H}_k)/\mathrm{tr}(\mathbf{H}_k)$ and the technical details are showed in appendix. Based on Lemma 3.16, one can show that our algorithms enjoy the local superlinear convergence for the general saddle point problems.

**Theorem 3.17.** *Solving general saddle point problem* (1) *under Assumption 2.1 and 2.2 by proposed random quasi-Newton methods (Algorithm 5, 6 and 7) by* $\mathbf{G}_0 = L^2\mathbf{I}$ *and* $M = 2\varkappa^2 L_2/L$, *we have the following results for all* $k > 0$:

1. *For the random Broyden family method (Algorithm 5), if* $\lambda_0$ *satisfies that* $\frac{M\lambda_0}{\mu} \leq \frac{\ln 2}{8(1+2d)\varkappa^2}$, *we have* $\mathbb{E}\left[\lambda_{k+1}/\lambda_k\right] \leq \left(1 - 1/(d\varkappa^2)\right)^k 2d\varkappa^2$.

2. *For the random BFGS method (Algorithm 6), if* $\lambda_0$ *satisfies that* $\frac{M\lambda_0}{\mu} \leq \frac{\ln 2}{8(1+d)\varkappa^2}$, *we have* $\mathbb{E}\left[\lambda_{k+1}/\lambda_k\right] \leq \left(1 - 1/d\right)^k 2d\varkappa^2$.

3. *For the random SR1 method (Algorithm 7), if* $\lambda_0$ *satisfies that* $\frac{M\lambda_0}{\mu} \leq \frac{\ln 2}{8(1+2d\varkappa^2)\varkappa^2}$, *we have* $\mathbb{E}\left[\lambda_{k+1}/\lambda_k\right] \leq \left(1 - 1/d\right)^k 2d\varkappa^4$.

Finally, combining the results of Theorem 3.15 and 3.17, we can prove the algorithms achieve the two-stages convergence behaviors as follows.

**Corollary 3.18.** *Solving the general saddle point problem* (1) *under Assumption 2.1 and 2.2 by our random quasi-Newton methods (Algorithm 5, 6 and 7) with* $\mathbf{G}_0 = L^2\mathbf{I}$ *and* $M = 2\varkappa^2 L_2/L$, *if the initial point is sufficiently close to the saddle point such that* $M\lambda_0/\mu \leq \ln 2/(8\varkappa^2)$, *then with probability* $1 - \delta$ *for any* $\delta \in (0, 1)$, *we have the following results:*

1. *The random Broyden family method (Algorithm 5) holds that*

$$\lambda_{k_0+k} \leq \left(1 - \frac{1}{d\varkappa^2 + 1}\right)^{\frac{k(k-1)}{2}} \left(\frac{1}{2}\right)^k \left(1 - \frac{1}{4\varkappa^2}\right)^{k_0} \lambda_0$$

*for all* $k_0 = \mathcal{O}\left(d\varkappa^2 \ln(d\varkappa/\delta)\right)$ *and* $k \geq 0$.

2. *The random BFGS/SR1 method (Algorithm 6/7) holds that*

$$\lambda_{k_0+k} \leq \left(1 - \frac{1}{d + 1}\right)^{\frac{k(k-1)}{2}} \left(\frac{1}{2}\right)^k \left(1 - \frac{1}{4\varkappa^2}\right)^{k_0} \lambda_0$$

*for all* $k_0 = \mathcal{O}\left(\max\left\{d, \varkappa^2\right\} \ln(d\varkappa/\delta)\right)$ *and* $k \geq 0$.

# 4 Discussion

The relationship between our quasi-Newton methods (Algorithm 5, 6 and 7) and existing first-order method for minimax problem is similar to the one for minimization problem. For our BFGS and SR1 methods, the dependency on $\varkappa^2$ only appears on the first period of linear convergence, which matches the convergence rate of gradient descent ascent. As an analogy, minimizing strongly-convex function by quasi-Newton methods [20, 32] has dependency on $\varkappa$ in the first period of linear convergence, which matches the convergence rate of gradient descent.

Our quasi-Newton methods also can be extended for solving non-linear equations, and the two-stage convergence behavior still hold. We provide the detailed discussion and the comparison with related work [19, 41] in Appendix F.

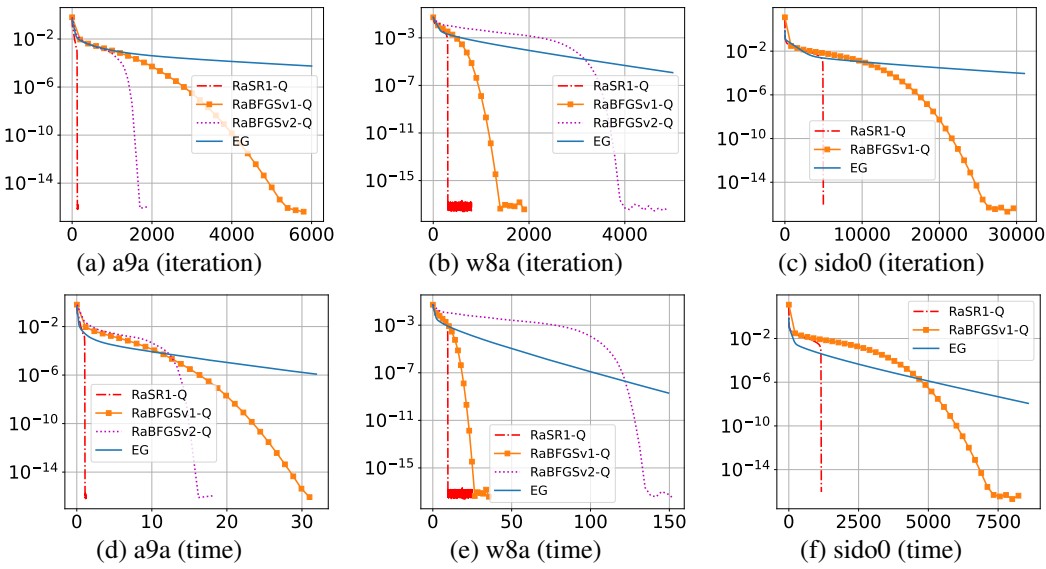

Figure 1: We demonstrate iteration numbers vs. $\|\mathbf{g}(\mathbf{z})\|_2$ and CPU time (second) vs. $\|\mathbf{g}(\mathbf{z})\|_2$ for AUC model on datasets "a9a" ($d = 126$, $n = 32561$), "w8a" ($d = 303$, $n = 45546$) and "sido" ($d = 4935$, $n = 12678$).

## 5 Numerical Experiments

In this section, we conduct the experiments on machine learning applications of AUC Maximization and adversarial debiasing to verify our theory. We refer to Algorithm 2 and 5 with parameter $\tau_k = \mathbf{u}^\top \mathbf{H}_{k+1}/(\mathbf{u}^\top \mathbf{G}_k \mathbf{u})$ as RaBFGSv1-Q and RaBFGSv1-G; refer to Algorithm 3, 4, 6 and 7 as RaBFGSv2-Q, RaSR1-Q, RaBFGSv2-G and RaSR1-G respectively. We compare these proposed algorithms with classical first-order method extragradient (EG) [18, 38].

### 5.1 AUC Maximization

AUC maximization [15, 43] aims to find the classifier $\mathbf{w} \in \mathbb{R}^m$ on the training set $\{\mathbf{a}_i, b_i\}_{i=1}^n$ where $\mathbf{a}_i \in \mathbb{R}^m$ and $b_i \in \{+1, -1\}$. We denote $n^+$ be the number of positive instances and $p = n^+/n$. The minimax formulation for AUC maximization can be written as

$$\min_{\mathbf{x} \in \mathbb{R}^{m+2}} \max_{y \in \mathbb{R}} f(\mathbf{x}, y) \stackrel{\text{def}}{=} \frac{1}{n} \sum_{i=1}^n f_i(\mathbf{x}, \mathbf{y}; \mathbf{a}_i, b_i, \lambda),$$

where $\mathbf{x} = [\mathbf{w}; u; v] \in \mathbb{R}^{m+2}$, $\lambda > 0$ is the regularization parameter and $f_i$ is defined as

$$
\begin{aligned}
f_i(\mathbf{x}, \mathbf{y}; \mathbf{a}_i, b_i, \lambda) =& \frac{\lambda}{2}\|\mathbf{x}\|_2^2 - p(1-p)y^2 + p((\mathbf{w}^T \mathbf{a}_i - v)^2 + 2(1+y)\mathbf{w}^T\mathbf{a}_i)\mathbb{I}_{b_i=-1} \\
&+ (1-p)((\mathbf{w}^T\mathbf{a}_i - u)^2 - 2(1+y)\mathbf{w}^T\mathbf{a}_i)\mathbb{I}_{b_i=1}.
\end{aligned}
$$

The objective function of AUC maximization is quadratic, hence we conduct the algorithms in Section 3.2 (Algorithm 2, 3 and 4) for this model. We set $\lambda = 100/n$ and evaluate all algorithms on three real-world datasets "a9a", "w8a" and "sido0". The dimension of the problem is $d = m + 3$. The results of iteration numbers against $\|\mathbf{g}(\mathbf{z})\|_2$ and CPU time against $\|\mathbf{g}(\mathbf{z})\|_2$ are presented in Figure 1. These results show that our algorithms perform better than the EG method.

### 5.2 Adversarial Debiasing

Adversarial learning [22, 44] can be used on fairness-aware machine learning issues. Give the training set $\{\mathbf{a}_i, b_i, c_i\}_{i=1}^n$, where $\mathbf{a}_i \in \mathbb{R}^d$ contains all input variables, $b_i \in \mathbb{R}$ is the output and $c_i \in \mathbb{R}$ is the input variable which we want to protect and make it unbiased. Our experiments are based on the

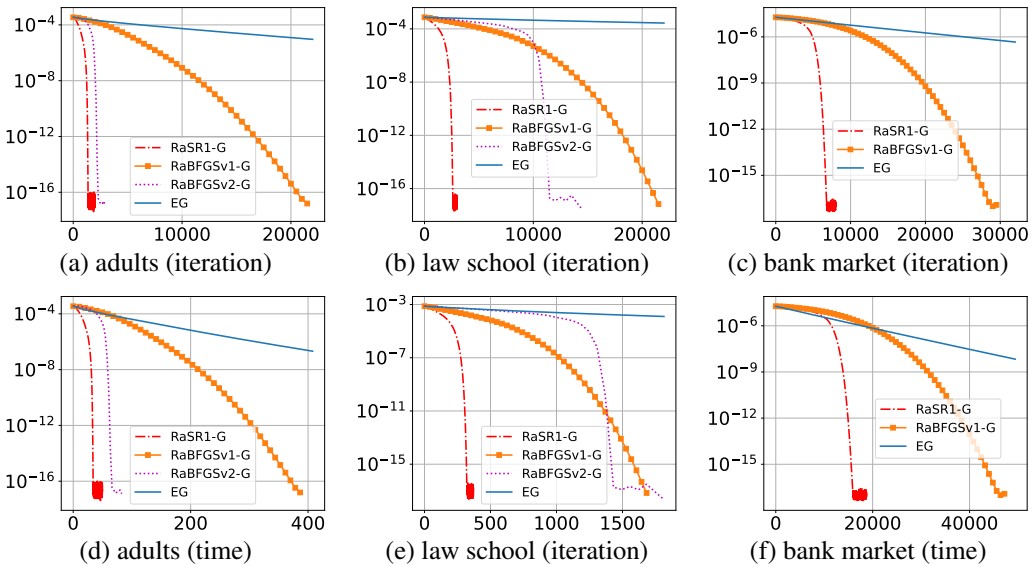

Figure 2: We demonstrate iteration numbers vs. $\|\mathbf{g}(\mathbf{z})\|_2$ and CPU time (second) vs. $\|\mathbf{g}(\mathbf{z})\|_2$ for adversarial debiasing model on datasets "adults" ($d = 123$, $n = 32561$), "law school" ($d = 380$, $n = 20427$) and "bank market" ($d = 3880$, $n = 45211$).

fairness-aware binary classification dataset "adult", "bank market" and "law school"[30], leading to $b_i, c_i \in \{+1, -1\}$. The model is formulated by the minimax problem

$$\min_{\mathbf{x} \in \mathbb{R}^m} \max_{y \in \mathbb{R}} \frac{1}{n} \sum_{i=1}^{n} (l_1(\mathbf{a}_i, b_i, \mathbf{x}) - \beta l_2(\mathbf{a}_i^\top \mathbf{x}, c_i, y)) + \lambda \|\mathbf{x}\|^2 - \gamma y^2,$$

where $l_1, l_2$ are the logit functions: $\mathrm{logit}(\mathbf{a}, b, \mathbf{c}) = \log(1 + \exp(-b\mathbf{a}^\top \mathbf{c}))$. We set the parameters $\beta, \lambda$ and $\gamma$ as $0.5, 10^{-4}$ and $10^{-4}$ respectively. The dimension of the problem is $d = m + 1$. Since the objective function is non-quadratic, we conduct the proposed algorithms in Section 3.3 (Algorithm 5, 6 and 7) here. We use extragradient as warm up to achieve the local condition for proposed algorithms. The results of iteration numbers against $\|\mathbf{g}(\mathbf{z})\|_2$ and CPU time against $\|\mathbf{g}(\mathbf{z})\|_2$ are presented in Figure 2, which indicate that our algorithms significantly outperform the baseline algorithm.

## 6 Conclusion

In this work, we propose quasi-Newton methods for solving strongly-convex-strongly-concave saddle point problems. We characterize the second-order information by approximating the square of Hessian matrix, which avoids the issue of dealing with the indefinite Hessian directly. We present the explicit local superlinear convergence rates for Broyden's family update and faster convergence rates for two specific methods: SR1 and BFGS updates. However, our algorithms still require to compute the exact gradient and store the approximate Hessian by $\mathcal{O}(d^2)$ space complexity. In future work, it is interesting to design the stochastic algorithms to further reduce the computational cost of the iteration. It is also possible to design the limited-memory quasi-Newton methods for more scalable saddle point problems.

## Acknowledgements

We would like to thank the anonymous reviewer for pointing out the mistakes in our previous proof. We would also like to thank Tong Zhang and John C.S. Lui for giving useful suggestions. This work is supported by National Natural Science Foundation of China (No. 62206058) and Shanghai Sailing Program (22YF1402900).

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
