## A   Efficient Implementation for Algorithm 1 and Proof of Section 3.1

For the self-completeness of this paper, we present Proposition 1 of Lin et al. [24] to show Algorithm 1 can be implemented with $\mathcal{O}\left(d^2\right)$ flops.

**Lemma A.1** (Lin et al. [24, Proposition 1])**.** *In this Lemma, we show how to construct upper triangle matrix $\hat{\mathbf{L}}$ from $\mathbf{L}$, $\mathbf{H}$ and the direction $\mathbf{u}$ with $\mathcal{O}(d^2)$ flops. From the inverse BFGS update rule of $\mathbf{A} = \mathbf{G}^{-1}$, we have*

$$
\begin{aligned}
\mathbf{A}_+ &= \left(\mathrm{BFGS}\left(\mathbf{G}, \mathbf{H}, \mathbf{u}\right)\right)^{-1} \\
&= \left(\mathbf{I} - \frac{\mathbf{u}\mathbf{u}^\top\mathbf{H}}{\mathbf{u}^\top\mathbf{H}\mathbf{u}}\right) \mathbf{A} \left(\mathbf{I} - \frac{\mathbf{H}\mathbf{u}\mathbf{u}^\top}{\mathbf{u}^\top\mathbf{H}\mathbf{u}}\right) + \frac{\mathbf{u}\mathbf{u}^\top}{\mathbf{u}^\top\mathbf{H}\mathbf{u}}.
\end{aligned}
\tag{8}
$$

*Suppose we already have $\mathbf{A} = \mathbf{L}^\top\mathbf{L}$ where $\mathbf{L}$ is an upper triangular matrix, now we construct $\hat{\mathbf{L}}$ such that $\mathbf{A}_+ = \hat{\mathbf{L}}^\top\hat{\mathbf{L}}$ with $\mathcal{O}(d^2)$ flops.*

*1. First we can obtain $QR$ decomposition of*

$$
\mathbf{L}\left(\mathbf{I} - \frac{\mathbf{H}\mathbf{u}\mathbf{u}^\top}{\mathbf{u}^\top\mathbf{H}\mathbf{u}}\right) = \mathbf{L} - \frac{\mathbf{L}(\mathbf{H}\mathbf{u})}{\mathbf{u}^\top\mathbf{H}\mathbf{u}}\mathbf{u}^\top
$$

*with $\mathcal{O}(d^2)$ flops since it is a rank-one changes of $\mathbf{L}$.*

*2. Second, we have*

$$
\mathbf{L}\left(\mathbf{I} - \frac{\mathbf{H}\mathbf{u}\mathbf{u}^\top}{\mathbf{u}^\top\mathbf{H}\mathbf{u}}\right) = \mathbf{Q}\mathbf{R},
$$

*with an orthogonal matrix $\mathbf{Q} \in \mathbb{R}^{d \times d}$ and an upper triangular matrix $\mathbf{R} \in \mathbb{R}^{d \times d}$. Denote $\mathbf{v} = \mathbf{u}/\sqrt{\mathbf{u}^\top\mathbf{H}\mathbf{u}}$, then we have*

$$
\mathbf{A}_+ = \mathbf{R}^\top\mathbf{Q}^\top\mathbf{Q}\mathbf{R} + \frac{\mathbf{u}\mathbf{u}^\top}{\mathbf{u}^\top\mathbf{H}\mathbf{u}} = \mathbf{R}^\top\mathbf{R} + \mathbf{v}\mathbf{v}^\top = \begin{bmatrix} \mathbf{v} & \mathbf{R}^\top \end{bmatrix} \begin{bmatrix} \mathbf{v}^\top \\ \mathbf{R} \end{bmatrix}.
$$

*we still can obtain $QR$ decomposition of $\begin{bmatrix} \mathbf{v}^\top \\ \mathbf{R} \end{bmatrix}$ with only $\mathcal{O}(d^2)$ flops, leading to $\begin{bmatrix} \mathbf{v}^\top \\ \mathbf{R} \end{bmatrix} = \mathbf{Q}'\mathbf{R}'$, with an column orthogonal matrix $\mathbf{Q}' \in \mathbb{R}^{(d+1) \times d}$ and an upper triangular matrix $\mathbf{R}' \in \mathbb{R}^{d \times d}$, and*

$$
\mathbf{A}_+ = \mathbf{R}'^\top\mathbf{R}'.
$$

*Thus $\mathbf{R}'$ satisfies our requirements.*

### A.1   Proofs of Lemma 3.3, 3.4 and 3.6

*Proof.* The results of Lemma 3.3, 3.4 and 3.6 can be directly derived from Theorem 3.1, 4.2 and 4.1 of Lin et al. [24].

Take $k = 1$ in Theorem 3.1, 4.1 and 4.2 of Lin et al. [24] we have

$$
\mathbb{E}\left[\sigma_{\mathbf{H}}(\mathbf{G}_1)\right] \leq \left(1 - \frac{1}{d\kappa^2}\right)\sigma_{\mathbf{H}}(\mathbf{G}_0)
$$

$$
\mathbb{E}\left[\tau_{\mathbf{H}}(\mathbf{G}_1)\right] \leq \left(1 - \frac{1}{d}\right)\tau_{\mathbf{H}}(\mathbf{G}_0)
$$

$$
\mathbb{E}\left[\sigma_{\mathbf{H}}(\mathbf{G}_1)\right] \leq \left(1 - \frac{1}{d}\right)\sigma_{\mathbf{H}}(\mathbf{G}_0).
$$

Replacing $\mathbf{G}_1$ and $\mathbf{G}_0$ with $\mathbf{G}_+$ and $\mathbf{G}$ in the above inequalities, we obtain the results of Lemma 3.3, 3.6 and 3.4. $\qquad\square$

# B   Useful Lemmas for Convergence Analysis and Proof for Lemma 3.1

This section provides two useful lemmas of symmetric positive definite matrices for our further analysis.

**Lemma B.1** (Rodomanov and Nesterov [37, Lemma 2.1 and Lemma 2.2]). *Suppose two positive definite matrices* $\mathbf{H}, \mathbf{G} \in \mathbb{R}^{d \times d}$ *satisfy* $\mathbf{H} \preceq \mathbf{G} \preceq \eta \mathbf{H}$ *for some* $\eta \geq 1$*, then for any* $\mathbf{u} \in \mathbb{R}^d$ *and* $\tau_1 < \tau_2$*, we have*

$$\mathrm{Broyd}_{\tau_1}(\mathbf{G}, \mathbf{H}, \mathbf{u}) \preceq \mathrm{Broyd}_{\tau_2}(\mathbf{G}, \mathbf{H}, \mathbf{u}).$$

*Additionally, for any* $\tau \in [0, 1]$*, we have*

$$\mathbf{H} \preceq \mathrm{Broyd}_{\tau}(\mathbf{G}, \mathbf{H}, \mathbf{u}) \preceq \eta \mathbf{H}. \tag{9}$$

**Lemma B.2.** *Suppose* $\mathbf{H}, \mathbf{G} \in \mathbb{R}^{d \times d}$ *are symmetric positive definite matrices,* $\hat{\mathbf{H}} \in \mathbb{R}^{d \times d}$ *are symmetric non-degenerate matrix where* $\mathbf{H} = (\hat{\mathbf{H}})^2$ *and*

$$\mathbf{H} \preceq \mathbf{G} \preceq \eta \mathbf{H},$$

*where* $\eta > 1$*. Then we have*

$$\|\mathbf{I} - \hat{\mathbf{H}} \mathbf{G}^{-1} \hat{\mathbf{H}}\| \leq 1 - \frac{1}{\eta}. \tag{10}$$

*Proof.* We have the following inequality for $\mathbf{G}^{-1}$ and $\mathbf{H}^{-1}$

$$\frac{1}{\eta} \mathbf{H}^{-1} \preceq \mathbf{G}^{-1} \preceq \mathbf{H}^{-1},$$

which means

$$\mathbf{0} \preceq \mathbf{H}^{-1} - \mathbf{G}^{-1} \preceq \left(1 - \frac{1}{\eta}\right) \mathbf{H}^{-1}.$$

Thus we have

$$\mathbf{0} \preceq \hat{\mathbf{H}}(\mathbf{H}^{-1} - \mathbf{G}^{-1})\hat{\mathbf{H}} \preceq \left(1 - \frac{1}{\eta}\right) \hat{\mathbf{H}} \mathbf{H}^{-1} \hat{\mathbf{H}} \preceq \left(1 - \frac{1}{\eta}\right) \mathbf{I}.$$

So we have

$$\|\mathbf{I} - \hat{\mathbf{H}} \mathbf{G}^{-1} \hat{\mathbf{H}}\| = \|\hat{\mathbf{H}}(\mathbf{H}^{-1} - \mathbf{G}^{-1})\hat{\mathbf{H}}\| \leq \left(1 - \frac{1}{\eta}\right).$$

$\square$

## B.1   Proof of Lemma 3.1

*Proof.* We partition $\hat{\mathbf{H}}(\mathbf{z}) \in \mathbb{R}^{d \times d}$ as

$$\hat{\mathbf{H}}(\mathbf{z}) = \begin{bmatrix} \hat{\mathbf{H}}_{\mathbf{xx}}(\mathbf{z}) & \hat{\mathbf{H}}_{\mathbf{xy}}(\mathbf{z}) \\ \hat{\mathbf{H}}_{\mathbf{xy}}(\mathbf{z})^{\top} & \hat{\mathbf{H}}_{\mathbf{yy}}(\mathbf{z}), \end{bmatrix} \in \mathbb{R}^{d \times d}$$

where the sub-matrices $\hat{\mathbf{H}}_{\mathbf{xx}}(\mathbf{z}) \in \mathbb{R}^{d_x \times d_x}$, $\hat{\mathbf{H}}_{\mathbf{xy}}(\mathbf{z}) \in \mathbb{R}^{d_x \times d_y}$, $\hat{\mathbf{H}}_{\mathbf{yx}}(\mathbf{z}) \in \mathbb{R}^{d_y \times d_x}$ and $\hat{\mathbf{H}}_{\mathbf{yy}}(\mathbf{z}) \in \mathbb{R}^{d_y \times d_y}$ satisfy $\hat{\mathbf{H}}_{\mathbf{xx}}(\mathbf{z}) \succeq \mu \mathbf{I}$, $\hat{\mathbf{H}}_{\mathbf{yy}}(\mathbf{z}) \preceq -\mu \mathbf{I}$ and $\|\hat{\mathbf{H}}(\mathbf{z})\| \leq L$ for all $\mathbf{z} \in \mathbb{R}^n$ by Assumption 2.1 and 2.2. We denote

$$\mathbf{J} = \begin{bmatrix} \mathbf{I}_{d_x} & \mathbf{0} \\ \mathbf{0} & -\mathbf{I}_{d_y} \end{bmatrix} \in \mathbb{R}^{d \times d},$$

where $\mathbf{I}_{d_x}$ and $\mathbf{I}_{d_y}$ are $d_x \times d_x$ identity matrix and $d_y \times d_y$ identity matrix respectively. We can verified that $\mathbf{J}^{\top}\mathbf{J} = \mathbf{I}$ and

$$\mathbf{J}\hat{\mathbf{H}}(\mathbf{z}) = \begin{bmatrix} \hat{\mathbf{H}}_{\mathbf{xx}}(\mathbf{z}) & \hat{\mathbf{H}}_{\mathbf{xy}}(\mathbf{z}) \\ -\hat{\mathbf{H}}_{\mathbf{xy}}(\mathbf{z})^{\top} & -\hat{\mathbf{H}}_{\mathbf{yy}}(\mathbf{z}) \end{bmatrix}.$$

Thus we have

$$\mathbf{H}(\mathbf{z}) = \hat{\mathbf{H}}(\mathbf{z})^{\top}\hat{\mathbf{H}}(\mathbf{z}) = \hat{\mathbf{H}}^{\top}(\mathbf{z})(\mathbf{J}^{\top}\mathbf{J})\hat{\mathbf{H}}(\mathbf{z}) = (\mathbf{J}\hat{\mathbf{H}})^{\top}(\mathbf{J}\hat{\mathbf{H}}).$$

Following Lemma 4.5 of Abernethy et al. [1], we have

$$\mathbf{H}(\mathbf{z}) \succeq \mu^2 \mathbf{I} \succ \mathbf{0}.$$

Since $\|\hat{\mathbf{H}}\| \leq L$ and $\mathbf{H} \succ \mathbf{0}$, we have

$$\|\mathbf{H}(\mathbf{z})\| = \|\hat{\mathbf{H}}(\mathbf{z})^2\| \leq \|\hat{\mathbf{H}}(\mathbf{z})\|^2 \leq L^2$$

and

$$\mathbf{H}(\mathbf{z}) \preceq L^2 \mathbf{I}.$$

$\square$

## C The Proof Details for Section 3.2

This section give the detailed proofs for the results and the two-stage convergence results for quadratic case in Section 3.2.

### C.1 The Proof of Lemma 3.7

*Proof.* We have $\mathbf{g}_k = \mathbf{A}\mathbf{z}_k - b$ and $\mathbf{g}_{k+1} = \mathbf{A}\mathbf{z}_{k+1} - b$ where $\mathbf{A} = \hat{\mathbf{H}}$, which means

$$\mathbf{g}_{k+1} - \mathbf{g}_k = \mathbf{A}(\mathbf{z}_{k+1} - \mathbf{z}_k) = -\hat{\mathbf{H}}\mathbf{G}_k^{-1}\hat{\mathbf{H}}\mathbf{g}_k.$$

So we have

$$\lambda_{k+1} = \|(\mathbf{I} - \hat{\mathbf{H}}\mathbf{G}_k^{-1}\hat{\mathbf{H}})\mathbf{g}_k\| \leq \|\mathbf{I} - \hat{\mathbf{H}}\mathbf{G}_k^{-1}\hat{\mathbf{H}}\|\lambda_k.$$

Since $\mathbf{H} \preceq \mathbf{G}_k \preceq \eta_k\mathbf{H}$ and $\mathbf{H} = (\hat{\mathbf{H}})^2$, According to Lemma B.2, we have $\|\mathbf{I} - \hat{\mathbf{H}}\mathbf{G}_k^{-1}\hat{\mathbf{H}}\| \leq \left(1 - \frac{1}{\eta_k}\right)$, which means

$$\lambda_{k+1} \leq \left(1 - \frac{1}{\eta_k}\right)\lambda_k.$$

$\square$

### C.2 The Proof of Theorem 3.8

*Proof.* Lemma 3.1 means $\mu^2\mathbf{I} \preceq \mathbf{H} \preceq L^2\mathbf{I}$. Combining with $\mathbf{G}_0 = L^2\mathbf{I}$, we have $\mathbf{H} \preceq \mathbf{G}_0 \preceq \varkappa^2\mathbf{H}$. According to Lemma B.1, we achieve

$$\mathbf{H} \preceq \mathbf{G}_k \preceq \varkappa^2\mathbf{H}.$$

Applying Lemma 3.7, we obtain

$$\lambda_{k+1} \leq \left(1 - \frac{1}{\varkappa^2}\right)\lambda_k \tag{11}$$

for all $k \geq 0$ which leads to

$$\lambda_k \leq \left(1 - \frac{1}{\varkappa^2}\right)^k \lambda_0. \tag{12}$$

$\square$

### C.3 The Proof of Theorem 3.9

*Proof.* The proof is modified from Lin et al. [24, Theorem 4.4]. Denote $\eta_k = \|\mathbf{H}^{-1/2}\mathbf{G}_k\mathbf{H}^{-1/2}\| \geq 1$, then we have

$$\mathbf{H} \preceq \mathbf{G}_k \preceq \eta_k\mathbf{H}$$

and

$$1 - \frac{1}{\eta_k} \leq \eta_k - 1 \overset{*}{\leq} \sigma_{\mathbf{H}}(\mathbf{G}_k) \leq \frac{\text{tr}(\mathbf{G}_k - \mathbf{H})}{\mu^2} = \frac{\tau_{\mathbf{H}}(\mathbf{G}_k)}{\mu^2}. \tag{13}$$

Inequality $*$ comes from the fact that

$$\eta_k - 1 \leq \|\mathbf{H}^{-1/2}(\mathbf{G}_k - \mathbf{H})\mathbf{H}^{-1/2}\| \leq \mathrm{tr}\left(\mathbf{H}^{-1/2}(\mathbf{G}_k - \mathbf{H})\mathbf{H}^{-1/2}\right) = \sigma_k.$$

**BFGS method:** According to Lemma 3.4, we have:

$$\mathbb{E}\left[\sigma_{\mathbf{H}}(\mathbf{G}_{k+1})\right] \leq \left(1 - \frac{1}{d}\right)\mathbb{E}\left[\sigma_{\mathbf{H}}(\mathbf{G}_k)\right].$$

which implies

$$\mathbb{E}\left[\sigma_{\mathbf{H}}(\mathbf{G}_k)\right] \leq \left(1 - \frac{1}{d}\right)^k \sigma_{\mathbf{H}}(\mathbf{G}_0).$$

Thus we have

$$\mathbb{E}\left[(\eta_k - 1)\right] \overset{(13)}{\leq} \mathbb{E}\left[\sigma_{\mathbf{H}}(\mathbf{G}_k)\right] \leq \left(1 - \frac{1}{d}\right)^k \sigma_{\mathbf{H}}(\mathbf{G}_0). \tag{14}$$

The upper bound of $\sigma_{\mathbf{H}}(\mathbf{G}_0)$ means

$$\sigma_{\mathbf{H}}(\mathbf{G}_0) = \langle\mathbf{H}^{-1}, \mathbf{G}_0\rangle - d \leq \langle\mathbf{H}^{-1}, \varkappa^2\mathbf{H}\rangle - d = d\left(\varkappa^2 - 1\right) \leq d\varkappa^2. \tag{15}$$

Combining with Lemma 3.7, we have

$$\begin{aligned}\mathbb{E}\left[\frac{\lambda_{k+1}}{\lambda_k}\right] &\leq \mathbb{E}\left[1 - \frac{1}{\eta_k}\right] \leq \mathbb{E}\left[\eta_k - 1\right]\\ &\overset{(14)}{\leq} \left(1 - \frac{1}{d}\right)^k \sigma_{\mathbf{H}}(\mathbf{G}_0)\\ &\overset{(15)}{\leq} \left(1 - \frac{1}{d}\right)^k d\varkappa^2.\end{aligned}$$

**Broyden Family Method:** The proof for the Broyden family method is similar to the one of BFGS. From Lemma 3.3, we have

$$\mathbb{E}\left[\sigma_{\mathbf{H}}(\mathbf{G}_{k+1})\right] \leq \left(1 - \frac{1}{d\varkappa^2}\right)\mathbb{E}\,\sigma_{\mathbf{H}}(\mathbf{G}_k).$$

which means

$$\mathbb{E}\left[\sigma_{\mathbf{H}}(\mathbf{G}_k)\right] \leq \left(1 - \frac{1}{d\varkappa^2}\right)^k \sigma_{\mathbf{H}}(\mathbf{G}_0).$$

The upper bound bound of $\sigma_{\mathbf{H}}(\mathbf{G}_0)$ is the same with the BFGS which implies

$$\mathbb{E}\left[\frac{\lambda_{k+1}}{\lambda_k}\right] \leq \mathbb{E}\left[\eta_k - 1\right] \leq \left(1 - \frac{1}{d\varkappa^2}\right)^k \sigma_{\mathbf{H}}(\mathbf{G}_0) \overset{(15)}{\leq} \left(1 - \frac{1}{d\varkappa^2}\right)^k d\varkappa^2.$$

**SR1 Method:** Using Theorem 4.1 of Lin et al. [24], we have

$$\mathbb{E}\left[\tau_{\mathbf{H}}(\mathbf{G}_k)\right] \leq \left(1 - \frac{k}{d}\right)\tau_{\mathbf{H}}(\mathbf{G}_0).$$

The upper bound of $\tau_{\mathbf{H}}(\mathbf{G}_k)$ means

$$\tau_{\mathbf{H}}(\mathbf{G}_0) = \mathrm{tr}(\mathbf{G}_0 - \mathbf{H}) \leq (\varkappa^2 - 1)\mathrm{tr}(\mathbf{H}) \leq d\varkappa^2 L^2.$$

Combining with Lemma 3.7, we have

$$\mathbb{E}\left[\frac{\lambda_{k+1}}{\lambda_k}\right] \leq \mathbb{E}\left[\eta_k - 1\right] \overset{(13)}{\leq} \mathbb{E}\left[\frac{\tau_{\mathbf{H}}(\mathbf{G}_k)}{\mu^2}\right] \leq \left(1 - \frac{k}{d}\right)\frac{d\varkappa^2 L^2}{\mu^2} = \left(1 - \frac{k}{d}\right)d\varkappa^4.$$

$\square$

Combining the results of Theorem 3.8 and 3.9, we achieve the two-stages convergence behavior, that is, the algorithm has global linear convergence and local superlinear convergence. The formal description is summarized as follows.

**Corollary C.1.** *Solving simple quadratic (SQ) saddle point problem by proposed random quasi-Newton algorithms, then with probability $1 - \delta$ for any $\delta \in (0, 1)$, we have the following results:*

1. *Using the random Broyden family method (Algorithm 2), we have*

$$\lambda_{k_0+k} \leq \left(1 - \frac{1}{d\varkappa^2 + 1}\right)^{\frac{k(k-1)}{2}} \left(\frac{1}{2}\right)^k \left(1 - \frac{1}{\varkappa^2}\right)^{k_0} \lambda_0$$

   *for all $k > 0$ and $k_0 = \mathcal{O}\left(d\varkappa^2 \ln(d\varkappa/\delta)\right)$.*

2. *Using the random BFGS method (Algorithm 3), we have*

$$\lambda_{k_0+k} \leq \left(1 - \frac{1}{d+1}\right)^{\frac{k(k-1)}{2}} \left(\frac{1}{2}\right)^k \left(1 - \frac{1}{\varkappa^2}\right)^{k_0} \lambda_0 \tag{16}$$

   *for all $k > 0$ and $k_0 = \mathcal{O}\left(d \ln(d\varkappa/\delta)\right)$.*

3. *Using the random SR1 method (Algorithm 4), we have*

$$\lambda_{k+k_0} \leq \frac{(d - k_0 - k + 1)!}{(d - k_0 + 1)!} \left(\frac{1}{2(d - k_0)}\right)^k \left(1 - \frac{1}{\varkappa^2}\right)^{k_0} \lambda_0$$

   *for all $d - k_0 + 1 \geq k > 0$ and $k_0 = \left\lceil \left(1 - \delta/(4d^2(d+1)\varkappa^4)\right) d \right\rceil$.*

*Note that for random SR1 method, $\lambda_k$ decrease to 0 with probability $1 - \delta$ when $k + k_0 = d + 1$.*

*Proof.* The convergence behaviors of the random algorithms also have two stages, the first one is global linear convergence and the second one is local superlinear convergence. We consider the random variable $X_k = \lambda_{k+1}/\lambda_k$ for all $k \geq 0$ in the following derivation.

**Broyden Family Method:** Note that given $X_k \geq 0$, using Markov's inequality and Theorem 3.8, we have

$$\mathbb{P}\left(X_k \geq \frac{d\varkappa^2}{\epsilon} \left(1 - \frac{1}{d\varkappa^2}\right)^k\right) \leq \frac{\mathbb{E}\left[X_k\right]}{\frac{d\varkappa^2}{\epsilon}\left(1 - \frac{1}{d\varkappa^2}\right)^k} \leq \epsilon \tag{17}$$

for any $\epsilon > 0$. Choosing $\epsilon_k = \delta(1 - q)q^k$ for some positive $q < 1$, then we have

$$\mathbb{P}\left(X_k \geq \frac{d\varkappa^2}{\epsilon_k} \left(1 - \frac{1}{d\varkappa^2}\right)^k, \exists k \in \mathbb{N}\right) \leq \sum_{k=0}^{\infty} \mathbb{P}\left(X_k \geq \frac{d\varkappa^2}{\epsilon_k} \left(1 - \frac{1}{d\varkappa^2}\right)^k\right)$$

$$\overset{(17)}{\leq} \sum_{k=0}^{\infty} \epsilon_k = \sum_{k=0}^{\infty} \delta(1 - q)q^k = \delta.$$

Therefore, we obtain

$$X_k \leq \left(\frac{1 - \frac{1}{d\varkappa^2}}{q}\right)^k \cdot \frac{d\varkappa^2}{(1 - q)\delta}$$

for all $k \in \mathbb{N}$ with probability $1 - \delta$.

If we set $q = 1 - 1/(d^2\varkappa^4)$, then it holds that

$$X_k \leq \frac{d^3\varkappa^6}{\delta} \left(1 + \frac{1}{d\varkappa^2}\right)^{-k} = \frac{d^3\varkappa^6}{\delta} \left(1 - \frac{1}{d\varkappa^2 + 1}\right)^k.$$

for all $k \in \mathbb{N}$ probability $1 - \delta$.

Furthermore, it holds that

$$\frac{\lambda_{k+1}}{\lambda_k} \leq \frac{d^3\varkappa^6}{\delta} \left(1 - \frac{1}{d\varkappa^2 + 1}\right)^k \tag{18}$$

for all $k \in \mathbb{N}$ with probability $1 - \delta$.

Telescoping from $k$ to $0$ in Eq. (18), we get

$$\lambda_k = \lambda_0 \cdot \prod_{i=1}^{k} \frac{\lambda_i}{\lambda_{i-1}} \leq \lambda_0 \cdot \left(\frac{d^3 \varkappa^6}{\delta}\right)^k \prod_{i=1}^{k} \left(1 - \frac{1}{d\varkappa^2 + 1}\right)^{i-1}$$

$$= \left(\frac{d^3 \varkappa^6}{\delta}\right)^k \left(1 - \frac{1}{d\varkappa^2 + 1}\right)^{k(k-1)/2} \lambda_0.$$

In the view of (18), we denote $k_0 \geq 0$ as the number of the first iteration satisfying

$$\frac{d^3 \varkappa^6}{\delta} \left(1 - \frac{1}{d\varkappa^2 + 1}\right)^{k_0} \leq \frac{1}{2}.$$

Clearly, we have $k_0 \leq (d\varkappa^2 + 1)\ln(2d^3\varkappa^6/\delta)$. Thus for all $k \geq 0$, we have

$$\lambda_{k_0 + k + 1} \overset{(18)}{\leq} \frac{d^3 \varkappa^6}{\delta} \left(1 - \frac{1}{d\varkappa^2 + 1}\right)^{k_0 + k} \lambda_{k_0 + k} \leq \frac{1}{2}\left(1 - \frac{1}{d\varkappa^2 + 1}\right)^k \lambda_{k_0 + k}.$$

Therefore, it holds that

$$\lambda_{k_0 + k} \leq \left(1 - \frac{1}{d\varkappa^2 + 1}\right)^{k(k-1)/2} \left(\frac{1}{2}\right)^k \lambda_{k_0},$$

and by Theorem 3.8 we have

$$\lambda_{k_0} \leq \left(1 - \frac{1}{\varkappa^2}\right)^{k_0} \lambda_0.$$

Finally, choose $k_0 = \mathcal{O}\left(d\varkappa^2 \ln(d\varkappa/\delta)\right)$, we obtain

$$\lambda_{k_0 + k} \leq \left(1 - \frac{1}{d\varkappa^2 + 1}\right)^{k(k-1)/2} \cdot \left(\frac{1}{2}\right)^k \cdot \left(1 - \frac{1}{\varkappa^2}\right)^{k_0} \lambda_0.$$

**BFGS Method:** Similar to the analysis for the random Broyden family method, we obtain with probability $1 - \delta$,

$$X_k \leq \left(\frac{1 - \frac{1}{d}}{q}\right)^k \cdot \frac{d\varkappa^2}{(1-q)\delta}, \qquad \text{for all } k \in \mathbb{N}.$$

If we set $q = 1 - 1/d^2$, we could obtain with probability $1 - \delta$,

$$\lambda_{k+1} \leq \frac{d^3 \varkappa^2}{\delta} \left(1 - \frac{1}{d+1}\right)^k \lambda_k, \qquad \text{for all } k \in \mathbb{N}. \tag{19}$$

We require the point $\mathbf{z}_{k_0}$ satisfies

$$\frac{2d^3 \varkappa^2}{\delta} \left(1 - \frac{1}{d+1}\right)^{k_0} \leq \frac{1}{2},$$

which can be guaranteed by setting $k_0 = \mathcal{O}\left(d\ln(d\varkappa/\delta)\right)$.

The remainder of the proof can follow the analysis in the random Broyden methods. We only need to replace all the term of $\left(1 - 1/(d\varkappa^2 + 1)\right)$ to $\left(1 - 1/(d+1)\right)$. The reason is that (19) provides a faster convergence result for the random BFGS update, rather than (18) for random Broyden method.

**SR1 Method:** Similar to the analysis for random Broyden family method, we obtain with probability $1 - \delta$,

$$X_k \leq \left(\frac{1 - \frac{k}{d}}{q}\right) \cdot \frac{d\varkappa^4}{(1-q)\delta} \qquad \text{for all } k \in \mathbb{N}.$$

If we set $q = 1 - 1/d^2$, we could obtain with probability $1 - \delta$,

$$\lambda_{k+1} \leq \frac{d^2(d+1)\varkappa^4}{\delta} \left(1 - \frac{k}{d}\right) \lambda_k \qquad \text{for all } 0 \leq k \leq d. \tag{20}$$

Recall that we denote $k_0$ the first iteration such that

$$\left(1 - \frac{k_0}{d}\right) \frac{d^2(d+1)\varkappa^4}{\delta} \le \frac{1}{2}.$$

Clearly, we can set

$$k_0 = \left\lceil \left(1 - \frac{\delta}{2d^2(d+1)\varkappa^4}\right) d \right\rceil.$$

Then it holds that

$$
\begin{aligned}
\lambda_{k_0+k+1} &\overset{(20)}{\le} \frac{d^2(d+1)\varkappa^4}{\delta}\left(1 - \frac{k+k_0}{d}\right) \\
&= \left(\frac{d-k-k_0}{d-k_0}\right)\left(1 - \frac{k_0}{d}\right)\frac{d^2(d+1)\varkappa^4}{\delta} \\
&\le \frac{1}{2}\left(\frac{d-k-k_0}{d-k_0}\right)\lambda_{k_0+k}.
\end{aligned}
$$

Thus for $k_0 = \left\lceil \left(1 - \delta/(2d^2(d+1)\varkappa^4)\right) d \right\rceil$ and $0 < k \le d - k_0 + 1$, we have

$$
\begin{aligned}
\lambda_{k+k_0} &\le \left(\frac{d-k+1-k_0}{d-k_0} \cdots \frac{d-k_0-1}{d-k_0}\right)\left(\frac{1}{2}\right)^k \lambda_{k_0} \\
&\le \frac{(d-k_0-k+1)!}{(d-k_0+1)!}\left(\frac{1}{2(d-k_0)}\right)^k \left(1 - \frac{1}{\varkappa^2}\right)^{k_0} \lambda_0
\end{aligned}
$$

$\square$

# D    The Proof Details of Section 3.3

This section give the detailed proofs for the results in Section 3.3.

## D.1    The Proof of Lemma 3.10

*Proof.* Assumption 2.1 mean operator $\hat{\mathbf{H}}(\cdot)$ is $L_2$-Lipschitz continuous and $\|\hat{\mathbf{H}}(\mathbf{z})\| \le L$ for all $\mathbf{z} \in \mathbb{R}^d$ then we have

$$
\begin{aligned}
\|\mathbf{H}(\mathbf{z}) - \mathbf{H}(\mathbf{z}')\| &= \|\hat{\mathbf{H}}(\mathbf{z})\hat{\mathbf{H}}(\mathbf{z}) - \hat{\mathbf{H}}(\mathbf{z}')\hat{\mathbf{H}}(\mathbf{z}')\| \\
&\le \|\hat{\mathbf{H}}(\mathbf{z})(\hat{\mathbf{H}}(\mathbf{z}) - \hat{\mathbf{H}}(\mathbf{z}'))\| + \|(\hat{\mathbf{H}}(\mathbf{z}) - \hat{\mathbf{H}}(\mathbf{z}'))\hat{\mathbf{H}}(\mathbf{z}')\| \\
&\le 2\|\hat{\mathbf{H}}\| \cdot \|\hat{\mathbf{H}}(\mathbf{z}) - \hat{\mathbf{H}}(\mathbf{z}')\| \\
&\le 2L_2 L\|\mathbf{z} - \mathbf{z}'\|.
\end{aligned}
$$

$\square$

## D.2    The Proof of Lemma 3.11

*Proof.* Lemma 3.1 shows that

$$\mu^2\mathbf{I} \preceq \mathbf{H}(\mathbf{z}) \preceq L^2\mathbf{I} \tag{21}$$

and Lemma 3.10 implies that

$$\|\mathbf{H}(\mathbf{z}) - \mathbf{H}(\mathbf{z}')\| \le 2L_2 L\|\mathbf{z} - \mathbf{z}'\|. \tag{22}$$

Combining the above lemmas, we have

$$
\begin{aligned}
\mathbf{H}(\mathbf{z}) - \mathbf{H}(\mathbf{z}') &\overset{(22)}{\preceq} 2L_2 L\|\mathbf{z} - \mathbf{z}'\|\mathbf{I} \\
&\overset{(21)}{\preceq} \frac{2L_2 L}{\mu^2}\|\mathbf{z} - \mathbf{z}'\|\mathbf{H}(\mathbf{w}) \\
&= \frac{2\varkappa^2 L_2}{L}\|\mathbf{z} - \mathbf{z}'\|\mathbf{H}(\mathbf{w}).
\end{aligned}
$$

$\square$

### D.3 The Proof of Corollary 3.12

*Proof.* According to Lemma 3.11, we have

$$\mathbf{H}(\mathbf{z}) - \mathbf{H}(\mathbf{z}') \preceq M\|\mathbf{z} - \mathbf{z}'\|\mathbf{H}(\mathbf{w}) \tag{23}$$

Taking interchanging of $\mathbf{z}'$ and $\mathbf{z}$ and letting $\mathbf{w} = \mathbf{z}$ in (23), we have

$$\mathbf{H}(\mathbf{z}') - \mathbf{H}(\mathbf{z}) \preceq M\|\mathbf{z} - \mathbf{z}'\|\mathbf{H}(\mathbf{z}) = Mr\mathbf{H}(\mathbf{z}).$$

Then taking $\mathbf{w} = \mathbf{z}'$, we have

$$\mathbf{H}(\mathbf{z}) - \mathbf{H}(\mathbf{z}') \preceq M\|\mathbf{z} - \mathbf{z}'\|\mathbf{H}(\mathbf{z}') = Mr\mathbf{H}(\mathbf{z}').$$

Combining above results we have

$$\frac{\mathbf{H}(\mathbf{z})}{1 + Mr} \preceq \mathbf{H}(\mathbf{z}') \preceq (1 + Mr)\mathbf{H}(\mathbf{z}), \tag{24}$$

. $\qquad\qquad\qquad\qquad\qquad\qquad\qquad\qquad\qquad\qquad\qquad\qquad\qquad\qquad\quad\square$

### D.4 The Proof of Lemma 3.13

*Proof.* By the conditiod, we have

$$\mathbf{H}(\mathbf{z}) \preceq \mathbf{G} \preceq \eta\mathbf{H}(\mathbf{z}), \tag{25}$$

Combining with Corollary 3.12 we have

$$\mathbf{H}(\mathbf{z}_+) \overset{(24)}{\preceq} (1 + Mr)\mathbf{H}(\mathbf{z}) \overset{(25)}{\preceq} (1 + Mr)\mathbf{G} = \tilde{\mathbf{G}} \tag{26}$$

and

$$\tilde{\mathbf{G}} = (1 + Mr)\mathbf{G} \overset{(25)}{\preceq} (1 + Mr)\eta\mathbf{H}(\mathbf{z}) \overset{(24)}{\preceq} (1 + Mr)^2\eta\mathbf{H}(\mathbf{z}_+),$$

which means

$$\mathbf{H}(\mathbf{z}_+) \preceq \tilde{\mathbf{G}} \preceq (1 + Mr)^2\eta\mathbf{H}(\mathbf{z}_+).$$

Then we obtain

$$\mathbf{H}(\mathbf{z}_+) \preceq \mathrm{Broyd}_\tau(\tilde{\mathbf{G}}, \mathbf{H}(\mathbf{z}_+), \mathbf{u}) \preceq (1 + Mr)^2\eta\mathbf{H}(\mathbf{z}_+). \tag{27}$$

$$\qquad\qquad\qquad\qquad\qquad\qquad\qquad\qquad\qquad\qquad\qquad\qquad\qquad\qquad\quad\square$$

### D.5 The Proof of Lemma 3.14

*Proof.* Recall that the objective satisfies the Assumption 2.1, that is

$$\|\mathbf{g}(\mathbf{z}_k) - \mathbf{g}(\mathbf{z}_{k+1})\| \leq L\|\mathbf{z}_k - \mathbf{z}_{k+1}\| \quad \text{and} \quad \|\hat{\mathbf{H}}(\mathbf{z}_k) - \hat{\mathbf{H}}(\mathbf{z}_{k+1})\| \leq L_2\|\mathbf{z}_k - \mathbf{z}_{k+1}\|. \tag{28}$$

We rewrite $\nabla f(\mathbf{z}_{k+1}) - \nabla f(\mathbf{z}_k)$ below

$$\nabla f(\mathbf{z}_{k+1}) - \nabla f(\mathbf{z}_k) = \int_0^1 \nabla^2 f(\mathbf{z}_k + s(\mathbf{z}_{k+1} - \mathbf{z}_k))(\mathbf{z}_{k+1} - \mathbf{z}_k)ds$$

$$= \int_0^1 \left[\nabla^2 f(\mathbf{z}_k + s(\mathbf{z}_{k+1} - \mathbf{z}_k)) - \nabla^2 f(\mathbf{z}_k)\right](\mathbf{z}_{k+1} - \mathbf{z}_k)ds + \nabla^2 f(\mathbf{z}_k)(\mathbf{z}_{k+1} - \mathbf{z}_k)$$

$$= \int_0^1 \left[\nabla^2 f(\mathbf{z}_k + s(\mathbf{z}_{k+1} - \mathbf{z}_k)) - \nabla^2 f(\mathbf{z}_k)\right](\mathbf{z}_{k+1} - \mathbf{z}_k)ds - \nabla^2 f(\mathbf{z}_k)\mathbf{G}_k^{-1}\nabla^2 f(\mathbf{z}_k)\nabla f(\mathbf{z}_k),$$

which means

$$\mathbf{g}_{k+1} = \underbrace{(\mathbf{I} - \hat{\mathbf{H}}_k\mathbf{G}_k^{-1}\hat{\mathbf{H}}_k)\mathbf{g}_k}_{\mathbf{a}_k} + \underbrace{\int_0^1 \left[\hat{\mathbf{H}}(\mathbf{z}_k + s(\mathbf{z}_{k+1} - \mathbf{z}_k)) - \hat{\mathbf{H}}(\mathbf{z}_k)\right](\mathbf{z}_{k+1} - \mathbf{z}_k)ds}_{\mathbf{b}_k}.$$

We first bound the term $\|\mathbf{a}_k\|$ by Lemma 3.13

$$\|\mathbf{a}_k\| \le \|\mathbf{I} - \hat{\mathbf{H}}_k \mathbf{G}_k^{-1} \hat{\mathbf{H}}_k\| \|\mathbf{g}_k\| \le \left(1 - \frac{1}{\eta_k}\right)\lambda_k. \tag{29}$$

Before we bound $\mathbf{b}_k$, we first try to bound $\hat{\mathbf{H}}_k \mathbf{G}_k^{-2} \hat{\mathbf{H}}_k$

$$\hat{\mathbf{H}}_k \mathbf{G}_k^{-2} \hat{\mathbf{H}}_k = (\hat{\mathbf{H}}_k \mathbf{G}_k^{-1/2}) \mathbf{G}_k^{-1} (\mathbf{G}_k^{-1/2} \hat{\mathbf{H}}_k) \preceq (\hat{\mathbf{H}}_k \mathbf{G}_k^{-1/2}) \frac{1}{\mu^2} \mathbf{I} (\mathbf{G}_k^{-1/2} \hat{\mathbf{H}}_k) = \frac{1}{\mu^2} \hat{\mathbf{H}}_k \mathbf{G}_k^{-1} \hat{\mathbf{H}}_k$$

$$\preceq \frac{1}{\mu^2} \hat{\mathbf{H}}_k \mathbf{H}_k^{-1} \hat{\mathbf{H}}_k \preceq \frac{1}{\mu^2} \mathbf{I}.$$

And $\|\mathbf{b}_k\|$ can be bounded by the $L_2$-Lipschitz continuity of the objective function

$$\|\mathbf{b}_k\| \le \int_0^1 \left\| \left(\hat{\mathbf{H}}\left(\mathbf{z}_k + s(\mathbf{z}_{k+1} - \mathbf{z}_k)\right) - \hat{\mathbf{H}}(\mathbf{z}_k)\right)(\mathbf{z}_{k+1} - \mathbf{z}_k)\right\| ds$$

$$\overset{(28)}{\le} \int_0^1 \|L_2 s(\mathbf{z}_{k+1} - \mathbf{z}_k)\| \|\mathbf{z}_{k+1} - \mathbf{z}_k\| ds \le \frac{L_2}{2} \|\mathbf{z}_{k+1} - \mathbf{z}_k\|^2$$

$$= \frac{L_2}{2} \|\mathbf{G}_k^{-1} \hat{\mathbf{H}}_k \mathbf{g}_k\|^2 = \frac{L_2}{2} \langle \mathbf{g}_k, \hat{\mathbf{H}}_k \mathbf{G}_k^{-2} \hat{\mathbf{H}}_k \mathbf{g}_k \rangle \le \frac{L_2}{2\mu^2} \|\mathbf{g}_k\|^2 = \beta \lambda_k^2.$$

Combining the above results, we have

$$\lambda_{k+1} \le \|\mathbf{a}_k\| + \|\mathbf{b}_k\| \le \left(1 - \frac{1}{\eta_k}\right)\lambda_k + \beta\lambda_k^2 \tag{30}$$

The relation of $r_k$ and $\lambda_k$ can be directly prove by the update formula

$$r_k = \|\mathbf{z}_{k+1} - \mathbf{z}_k\| = \|\mathbf{G}_k^{-1} \hat{\mathbf{H}}_k \mathbf{g}_k\| = \langle \mathbf{g}_k, \hat{\mathbf{H}}_k \mathbf{G}_k^{-2} \hat{\mathbf{H}}_k \mathbf{g}_k \rangle^{1/2} \le \frac{1}{\mu} \lambda_k \tag{31}$$

$\square$

### D.6 The proof of Theorem 3.15

*Proof.* We use induction to prove the following statements

$$\mathbf{H}_k \preceq \mathbf{G}_k \preceq \exp\left(2\sum_{i=0}^{k-1} \rho_i\right) \varkappa^2 \mathbf{H}_k \preceq b\varkappa^2 \mathbf{H}_k, \tag{32}$$

$$\lambda_k \le \left(1 - \frac{1}{2b\varkappa^2}\right)^k \lambda_0, \tag{33}$$

$$\eta_k \overset{\text{def}}{=} \exp\left(\sum_{i=0}^{k-1} 2\rho_i\right) \varkappa^2 \le b\varkappa^2 \tag{34}$$

hold for all $k \ge 0$. The initial assumption promise that $\lambda_0$ is small enough such that

$$\frac{M}{\mu}\lambda_0 \le \frac{\ln b}{4b\varkappa^2} \tag{35}$$

For $k = 0$, the initialization $\mathbf{G}_0 = L^2 \mathbf{I}$ leads to $\mathbf{H}_0 \preceq \mathbf{G}_0 \preceq \varkappa^2 \mathbf{H}_0$ and $\eta_0 = \varkappa^2$, which satisfy (32), (33) and (34). Then we prove these results for $k' = k + 1$.

The induction assumption means $\eta_k \le b\varkappa^2$ and $\mathbf{H}_k \preceq \mathbf{G}_k \le \eta_k \mathbf{H}_k$. Using Lemma 3.14, we have

$$\lambda_{k+1} \le \left(1 - \frac{1}{b\varkappa^2}\right)\lambda_k + \beta\lambda_k^2 \overset{(33)}{\le} \left(1 - \frac{1}{b\varkappa^2} + \beta\lambda_0\right)\lambda_k$$

$$\overset{(35)}{\le} \left(1 - \frac{1}{2b\varkappa^2}\right)\lambda_k \overset{(33)}{\le} \left(1 - \frac{1}{2b\varkappa^2}\right)^{k+1}\lambda_0.$$

Recall that we've defined $\rho_i = \frac{M\lambda_i}{\mu}$. Based on the fact $e^x \geq x + 1$ and Lemma 3.13, we have

$$
\begin{aligned}
\mathbf{H}_{k+1} \preceq \mathbf{G}_{k+1} &\overset{(26)}{\preceq} (1 + Mr_k)^2 \eta_k \mathbf{H}_{k+1} \preceq \left(1 + \frac{M\lambda_k}{\mu}\right)^2 \eta_k \mathbf{H}_{k+1} \\
&= (1 + \rho_k)^2 \eta_k \mathbf{H}_{k+1} \preceq e^{2\rho_k} \eta_k \mathbf{H}_{k+1} \\
&\overset{(32)}{\preceq} \exp\left(2\sum_{i=0}^{k} \rho_i\right) \varkappa^2 \mathbf{H}_{k+1};
\end{aligned}
$$

and the term $\sum_{i=0}^{k} \rho_i$ can be bounded by

$$
\begin{aligned}
\sum_{i=0}^{k} \rho_i &= \frac{M}{\mu} \sum_{i=0}^{k} \lambda_i \overset{(33)}{\leq} \frac{M}{\mu} \lambda_0 \sum_{i=0}^{k} \left(1 - \frac{1}{2b\varkappa^2}\right)^i \\
&\leq \frac{2bML^2}{\mu^3} \lambda_0 \overset{(35)}{\leq} \frac{\ln b}{2}.
\end{aligned}
\tag{36}
$$

Hence, for $k + 1$, we have

$$
\mathbf{G}_{k+1} \preceq \exp\left(2\sum_{i=1}^{k} \rho_i\right) \varkappa^2 \mathbf{H}_{k+1} \preceq b\varkappa^2 \mathbf{H}_{k+1} \quad \text{and} \quad \eta_{k+1} = \exp\left(2\sum_{i=1}^{k} \rho_i\right) \varkappa^2 \leq b\varkappa^2.
$$

Thus we have complete the proof for statements (32), (33) and (34) by induction. $\qquad \square$

### D.7 The Proof of Lemma 3.16

*Proof.* The proof of two algorithms are similar, the only difference is because two update formulas have different convergence rate. And from the Lemma 3.13, for both BFGS and Broyden family updates, we have

$$
\tilde{\mathbf{G}} \overset{\text{def}}{=} (1 + Mr)\mathbf{G} \succeq \mathbf{H}(\mathbf{z}).
\tag{37}
$$

We first give the proof for BFGS method.

**BFGS Method:** Using Lemma 3.4, we have

$$
\mathbb{E}_{\mathbf{u}_k}[\sigma_{k+1}] = \mathbb{E}_{\mathbf{u}_k}[\sigma_{\mathbf{H}_{k+1}}(\mathbf{G}_{k+1})] \leq \left(1 - \frac{1}{d}\right) \sigma_{\mathbf{H}_{k+1}}(\tilde{\mathbf{G}}_k).
$$

Then we bound the term $\sigma_{\mathbf{H}_{k+1}}(\tilde{\mathbf{G}}_k)$ by $\sigma_k$ as follows

$$
\begin{aligned}
\sigma_{\mathbf{H}_{k+1}}(\tilde{\mathbf{G}}_k) &= \langle \mathbf{H}_{k+1}^{-1}, \tilde{\mathbf{G}}_k \rangle - d \\
&\overset{(37)}{=} (1 + Mr_k) \langle \mathbf{H}_{k+1}^{-1}, \mathbf{G}_k \rangle - d \\
&\overset{(24)}{\leq} (1 + Mr_k)^2 \langle \mathbf{H}_k^{-1}, \mathbf{G}_k \rangle - d \\
&= (1 + Mr_k)^2 \sigma_k + d((1 + Mr_k)^2 - 1) \\
&= (1 + Mr_k)^2 \sigma_k + 2dMr_k \left(1 + \frac{Mr_k}{2}\right) \\
&\leq (1 + Mr_k)^2 \left(\sigma_k + \frac{2dMr_k}{1 + Mr_k}\right).
\end{aligned}
\tag{38}
$$

Combining above results, we obtain

$$
\mathbb{E}_{\mathbf{u}_k}[\sigma_{k+1}] \leq \left(1 - \frac{1}{d}\right)(1 + Mr_k)^2 \left(\sigma_k + \frac{2dMr_k}{1 + Mr_k}\right) \quad \text{for all } k \geq 0.
\tag{39}
$$

**Broyden Family Method:** If we use the Broyden family update

$$\mathbf{G}_{k+1} = \mathrm{Broyd}_{\tau_k}(\tilde{\mathbf{G}}_k, \mathbf{H}_{k+1}, \mathbf{u}_k)$$

instead of BFGS, Lemma 3.3 means

$$\mathbb{E}_{\mathbf{u}_k}[\sigma_{k+1}] \le \left(1 - \frac{1}{d\varkappa^2}\right) \sigma_{\mathbf{H}_{k+1}}(\tilde{\mathbf{G}}_k).$$

Combining with (38) which holds for both Broyden method and BFGS method, we obtain

$$\mathbb{E}_{\mathbf{u}_k}[\sigma_{k+1}] \le \left(1 - \frac{1}{d\varkappa^2}\right)(1 + Mr_k)^2 \left(\sigma_k + \frac{2dMr_k}{1 + Mr_k}\right) \quad \text{for all } k \ge 0. \tag{40}$$

$\square$

## D.8 The Proof of Theorem 3.17

We give the proof for the BFGS and Broyden Family methods in Section D.8.1 and the proof for the SR1 method in Section D.8.2. Taking $b = 2$, all algorithms have $\frac{M\lambda_0}{\mu} \le \frac{\ln 2}{8\varkappa^2}$, which implies the properties of $\lambda_k$ shown in Theorem 3.15.

Recall the definition of $\sigma_k$

$$\sigma_k \overset{\text{def}}{=} \sigma_{\mathbf{H}_k}(\mathbf{G}_k) = \langle \mathbf{H}_k^{-1}, \mathbf{G}_k \rangle - d. \tag{41}$$

If $\mathbf{G}_k \succeq \mathbf{H}_k$, then according to Rodomanov and Nesterov [37] it holds that

$$\mathbf{G}_k - \mathbf{H}_k \preceq \langle \mathbf{H}_k^{-1}, \mathbf{G}_k - \mathbf{H}_k \rangle \mathbf{H}_k = \sigma_k \mathbf{H}_k. \tag{42}$$

From Theorem 3.14, we have

$$\lambda_{k+1} \le \left(1 - \frac{1}{1 + \sigma_k}\right)\lambda_k + \beta\lambda_k^2, \tag{43}$$

and

$$r_k \le \frac{\lambda_k}{\mu} \tag{44}$$

for each $k \ge 0$.

### D.8.1 The Proofs of BFGS and Broyden Methods

**BFGS Method:** First we give the proof for the BFGS method. From Lemma 3.16, we obtain

$$\mathbb{E}_{\mathbf{u}_k}[\sigma_{k+1}] \le \left(1 - \frac{1}{d}\right)(1 + Mr_k)^2 \left(\sigma_k + \frac{2dMr_k}{1 + Mr_k}\right). \tag{45}$$

Since we have defined $\rho_k = M\lambda_k/\mu$ and the constant $\beta$, $M$ satisfy $\frac{\beta}{M} < \frac{1}{4L}$, it holds that

$$\rho_k \overset{(43)}{\le} \frac{\sigma_k}{1 + \sigma_k}\rho_k + \frac{\beta\mu}{2M}\rho_k^2 \le \sigma_k\rho_k + \frac{\beta\mu}{2M}\rho_k^2 < \sigma_k\rho_k + \frac{\mu}{8L}\rho_k^2 \le \sigma_k\rho_k + \frac{1}{8}\rho_k^2, \tag{46}$$

and

$$\mathbb{E}[\sigma_{k+1}] \overset{(45),(44)}{\le} \left(1 - \frac{1}{d}\right)\mathbb{E}\left[(1 + \rho_k)^2 \left(\sigma_k + \frac{2d\rho_k}{1 + \rho_k}\right)\right]. \tag{47}$$

We set

$$\theta_k \overset{\text{def}}{=} \sigma_k + 2d\rho_k$$

and consider Theorem 3.15 with $b = 2$. Then the convergence result of (36) and the initial assumption of $\mathbf{z}_0$ implies that

$$\rho_k \le \left(1 - \frac{1}{4\varkappa^2}\right)^k \rho_0, \tag{48}$$

and

$$\rho_0 \leq \frac{\ln 2}{8\varkappa^2(1+2d)}. \tag{49}$$

We now use induction to show that

$$\mathbb{E}\left[\theta_k\right] \leq \left(1 - \frac{1}{d}\right)^k 2d\varkappa^2. \tag{50}$$

In the case of $k = 0$, we have

$$\sigma_0 + 2d\rho_0 \overset{(41)}{=} \langle \mathbf{H}_0^{-1}, \mathbf{G}_0 \rangle - d + 2d\rho_0 \leq \langle \mathbf{H}_0^{-1}, \varkappa^2 \mathbf{H}_0 \rangle - d + 2d\rho_0 \tag{51}$$
$$= d\left(\varkappa^2 - 1\right) + 2d\rho_0 \leq d\varkappa^2.$$

Thus for $k = 0$, (50) is satisfied.

Suppose inequality (50) holds for $0 \leq k' \leq k$. For $k + 1$, using the inequality $e^x \geq 1 + x$, we have

$$\begin{aligned}
\mathbb{E}\left[\sigma_{k+1}\right] &\overset{(47)}{\leq} \left(1 - \frac{1}{d}\right) \mathbb{E}\left[(1 + \rho_k)^2 \left(\sigma_k + \frac{2d\rho_k}{1+\rho_k}\right)\right] \\
&\leq \left(1 - \frac{1}{d}\right) \mathbb{E}\left[(1 + \rho_k)^2(\sigma_k + 2d\rho_k)\right] \\
&= \left(1 - \frac{1}{d}\right) \mathbb{E}\left[(1 + \rho_k)^2 \theta_k\right] \\
&\leq \left(1 - \frac{1}{d}\right) \mathbb{E}\left[\exp\left(2\rho_k\right)\theta_k\right],
\end{aligned} \tag{52}$$

and

$$\rho_{k+1} \leq \rho_k \left(\sigma_k + \frac{1}{8}\rho_k\right) \leq \rho_k \left(\sigma_k + 2d\rho_k\right) \leq \left(1 - \frac{1}{d}\right) 2 \exp\left(2\rho_k\right)\theta_k\rho_k. \tag{53}$$

The last inequality of (53) comes from

$$2\left(1 - \frac{1}{d}\right) \exp(2\rho_k) \geq 1,$$

where we use the fact $\rho_k \geq 0$ and assume $n \geq 2$.

Thus we obtain

$$\begin{aligned}
\mathbb{E}\left[\sigma_{k+1} + 2d\rho_{k+1}\right] &\overset{(53),(52)}{\leq} \left(1 - \frac{1}{d}\right) \mathbb{E}\left[\exp\left(2\rho_k\right)\theta_k\right] + \left(1 - \frac{1}{d}\right) 4d\mathbb{E}\left[\exp\left(2\rho_k\right)\theta_k\rho_k\right] \\
&\leq \left(1 - \frac{1}{d}\right) \mathbb{E}\left[\exp\left(2\rho_k\right)\theta_k(1 + 4d\rho_k)\right] \\
&\leq \left(1 - \frac{1}{d}\right) \mathbb{E}\left[\exp\left(2\rho_k\right)\exp\left(4d\rho_k\right)\theta_k\right] \\
&\leq \left(1 - \frac{1}{d}\right) \mathbb{E}\left[\exp\left(2(1 + 2d)\rho_k\right)\theta_k\right] \\
&\overset{(48)}{\leq} \left(1 - \frac{1}{d}\right) \mathbb{E}\left[\exp\left(2(1 + 2d)\left(1 - \frac{1}{4\varkappa^2}\right)^k \rho_0\right)\theta_k\right] \\
&= \left(1 - \frac{1}{d}\right) \exp\left(2(1 + 2d)\left(1 - \frac{1}{4\varkappa^2}\right)^k \rho_0\right)\mathbb{E}\left[\theta_k\right].
\end{aligned}$$

Therefore, we have

$$\mathbb{E}\left[\theta_{k+1}\right] \quad \leq \quad \left(1 - \frac{1}{d}\right) \exp\left(2(1 + 2d)\left(1 - \frac{1}{4\varkappa^2}\right)^k \rho_0\right)\mathbb{E}\left[\theta_k\right]$$

$$\leq \quad \left(1 - \frac{1}{d}\right)^{k+1} \exp\left(2(1+2d)\rho_0 \sum_{i=0}^{k}\left(1 - \frac{1}{4\varkappa^2}\right)^i\right) \mathbb{E}\left[\theta_0\right]$$

$$\leq \quad \left(1 - \frac{1}{d}\right)^{k+1} \exp\left(8\varkappa^2(1+2d)\rho_0\right)\mathbb{E}\left[\theta_0\right]$$

$$\overset{(49),(51)}{\leq} \quad \left(1 - \frac{1}{d}\right)^{k+1} 2d\varkappa^2,$$

which proves (50). Hence, for any $k \geq 0$, we have

$$\mathbb{E}\left[\sigma_k\right] \leq \mathbb{E}\left[\theta_k\right] \leq \left(1 - \frac{1}{d}\right)^k 2d\varkappa^2,$$

which implies

$$\mathbb{E}\left[\frac{\lambda_{k+1}}{\lambda_k}\right] = \mathbb{E}\left[\frac{\rho_{k+1}}{\rho_k}\right] \overset{(53)}{\leq} \mathbb{E}\left[\sigma_k + 2d\rho_k\right] \leq \mathbb{E}\left[\theta_k\right] \leq \left(1 - \frac{1}{d}\right)^k 2d\varkappa^2. \tag{54}$$

**Broyden Family Method:** The proof for the Broyden method is almost the same as the one in the BFGS method. The reason that produce the different convergence result between Broyden and BGGS is that Lemma 3.16 only provides a slower convergence rate

$$\mathbb{E}[\sigma_{k+1}] \leq \left(1 - \frac{1}{d\varkappa^2}\right)(1 + Mr_k)^2\left(\sigma_k + \frac{2dMr_k}{1 + Mr_k}\right)$$

for Broyden method, rather than

$$\mathbb{E}[\sigma_{k+1}] \leq \left(1 - \frac{1}{d}\right)(1 + Mr_k)^2\left(\sigma_k + \frac{2dMr_k}{1 + Mr_k}\right)$$

for BFGS. Thus we can directly replace the term $\left(1 - \frac{1}{d}\right)$ to the term $\left(1 - \frac{1}{d\varkappa^2}\right)$ in the proof of BFGS method and obtain the convergence result of Broyden method

$$\mathbb{E}\left[\sigma_k\right] \leq \left(1 - \frac{1}{d\varkappa^2}\right)^k 2d\varkappa^2,$$

and

$$\mathbb{E}\left[\frac{\lambda_{k+1}}{\lambda_k}\right] \leq \left(1 - \frac{1}{d\varkappa^2}\right)^k 2d\varkappa^2.$$

### D.8.2 The Proof of SR1 Method

Note that we use $\tau_k$ to replace $\sigma_k$ for measuring how well does $\mathbf{G}_k$ approximate $\mathbf{H}_k$. We modify the proof in Lin et al. [24, Lemma 4.6] to obtain our results.

**SR1 Method:** First, we define the random sequence $\{\eta_k\}$ as follows

$$\eta_k \overset{\text{def}}{=} \frac{\text{tr}(\mathbf{G}_k - \mathbf{H}_k)}{\text{tr}(\mathbf{H}_k)}. \tag{55}$$

Since $\mathbf{G}_{k+1} = \text{SR1}(\widetilde{\mathbf{G}}_k, \mathbf{H}_{k+1}, \mathbf{u}_k)$ and according to Lemma 3.6 we have

$$
\begin{aligned}
\mathbb{E}_{\mathbf{u}_k}[\text{tr}(\mathbf{G}_{k+1} - \mathbf{H}_{k+1})] \quad &\leq \quad (1 - \frac{1}{d})\text{tr}(\widetilde{\mathbf{G}}_k - \mathbf{H}_{k+1}) \\
&\overset{(24)}{\leq} \quad \left(1 - \frac{1}{d}\right)\text{tr}\left((1 + Mr_k)\mathbf{G}_k - \frac{1}{1 + Mr_k}\mathbf{H}_k\right) \\
&\overset{(55)}{=} \quad \left(1 - \frac{1}{d}\right)\left((1 + Mr_k)(1 + \eta_k) - \frac{1}{1 + Mr_k}\right)\text{tr}(\mathbf{H}_k) \\
&\leq \quad \left(1 - \frac{1}{d}\right)\left((1 + Mr_k)^2(1 + \eta_k) - 1\right)\text{tr}(\mathbf{H}_{k+1}).
\end{aligned}
$$

Thus, we obtain

$$
\begin{aligned}
\mathbb{E}_{\mathbf{u}_k}\left[\eta_{k+1}\right] &\leq \left(1 - \frac{1}{d}\right)\left((1 + Mr_k)^2(1 + \eta_k) - 1\right) \\
&\leq \left(1 - \frac{1}{d}\right)\left[(1 + Mr_k)^2\eta_k + Mr_k(Mr_k + 2)\right] \\
&\leq \left(1 - \frac{1}{d}\right)(1 + Mr_k)^2(\eta_k + 2Mr_k).
\end{aligned}
\tag{56}
$$

The last inequality comes from the fact that

$$
t(t + 2) = t^2 + 2t \leq 2t + 4t^2 + 2t^3 = (1 + t^2)2t
$$

for all $t > 0$.

Since we have $\mu^2\mathbf{I} \preceq \mathbf{H}_k \preceq L^2\mathbf{I}$, then

$$
\sigma_k = \text{tr}((\mathbf{G}_k - \mathbf{H}_k)\mathbf{H}_k^{-1}) \leq \frac{1}{\mu^2}\text{tr}(\mathbf{G}_k - \mathbf{H}_k) = \frac{\eta_k}{\mu^2}\text{tr}(\mathbf{H}_k) \leq d\eta_k\varkappa^2.
$$

It also holds that

$$
\lambda_{k+1} \overset{(43)}{\leq} \left(1 - \frac{1}{1 + \sigma_k}\right)\lambda_k + \beta\lambda_k^2 \leq \sigma_k\lambda_k + \beta\lambda_k^2 \leq (d\varkappa^2\eta_k)\lambda_k + \beta\lambda_k^2.
$$

Recall that $\rho_k = M\lambda_k/\mu$, and we have

$$
\begin{aligned}
2\rho_{k+1} &\leq (2d\varkappa^2\eta_k)\rho_k + \frac{1}{4}\rho_k^2 \\
&\leq 2d\varkappa^2\rho_k(\eta_k + 2\rho_k) \\
&\leq \left(1 - \frac{1}{d}\right)4d\varkappa^2\rho_k(\eta_k + 2\rho_k) \\
&\leq \left(1 - \frac{1}{d}\right)(1 + \rho_k)^2 4d\varkappa^2\rho_k(\eta_k + 2\rho_k),
\end{aligned}
\tag{57}
$$
$$
\tag{58}
$$

and

$$
\mathbb{E}\left[\eta_{k+1}\right] \overset{(56)}{\leq} \left(1 - \frac{1}{d}\right)\mathbb{E}\left[(1 + \rho_k)^2(\eta_k + 2\rho_k)\right].
\tag{59}
$$

Combing above results, we obtain

$$
\begin{aligned}
\mathbb{E}\left[\eta_{k+1} + 2\rho_{k+1}\right] &\overset{(59),(58)}{\leq} \left(1 - \frac{1}{d}\right)\mathbb{E}\left[(1 + \rho_k)^2(1 + 4d\varkappa^2\rho_k)(\eta_k + 2\rho_k)\right] \\
&\leq \left(1 - \frac{1}{d}\right)\mathbb{E}\left[\exp\left(2\rho_k + 4d\varkappa^2\rho_k\right)(\eta_k + 2\rho_k)\right].
\end{aligned}
\tag{60}
$$

Let $\theta_k \overset{\text{def}}{=} \eta_k + 2\rho_k$, then

$$
\mathbb{E}\left[\theta_{k+1}\right] \leq \left(1 - \frac{1}{d}\right)\mathbb{E}\left[\exp\left(2(1 + 2d\varkappa^2)\rho_k\right)\theta_k\right].
$$

The initial condition means that

$$
\rho_0 \leq \frac{\ln 2}{8(1 + 2d\varkappa^2)\varkappa^2}.
\tag{61}
$$

In the following, we use induction to prove the fact that

$$
\mathbb{E}\left[\theta_k\right] \leq \left(1 - \frac{1}{d}\right)^k 2\varkappa^2.
$$

For $k = 0$, the initial condition $\mathbf{G}_0 \preceq \varkappa^2\mathbf{H}_0$ means $\eta_0 \leq \varkappa^2 - 1$. Thus we obtain

$$
\theta_0 = 2\rho_0 + \eta_0 \overset{(61)}{\leq} \frac{\ln 2}{4(1 + 2d\varkappa^2)\varkappa^2} + (\varkappa^2 - 1) \leq \varkappa^2.
\tag{62}
$$

For $k \geq 1$, we have

$$
\begin{aligned}
\mathbb{E}\left[\theta_{k+1}\right] & \stackrel{(60)}{\leq}\left(1-\frac{1}{d}\right) \mathbb{E}\left[\exp \left(2(1+2 d \varkappa^2) \rho_k\right) \theta_k\right] \\
& \stackrel{(48)}{\leq}\left(1-\frac{1}{d}\right) \mathbb{E}\left[\exp \left(2(1+2 d \varkappa^2)\left(1-\frac{1}{4 \varkappa^2}\right)^k \rho_0\right) \theta_k\right] \\
& =\left(1-\frac{1}{d}\right) \exp \left(2(1+2 d \varkappa^2)\left(1-\frac{1}{4 \varkappa^2}\right)^k \rho_0\right) \mathbb{E}\left[\theta_k\right] \\
& \leq\left(1-\frac{1}{d}\right)^{k+1} \exp \left(2(1+2 d \varkappa^2) \rho_0 \sum_{i=0}^k\left(1-\frac{1}{4 \varkappa^2}\right)^i\right) \mathbb{E}\left[\theta_0\right] \\
& \leq\left(1-\frac{1}{d}\right)^{k+1} \exp \left(8 \varkappa^2(1+2 d \varkappa^2) \rho_0\right) \mathbb{E}\left[\theta_0\right] \\
& \stackrel{(61),(62)}{\leq}\left(1-\frac{1}{d}\right)^{k+1} 2 \varkappa^2,
\end{aligned}
$$
(63)

which implies

$$
\mathbb{E}\left[\eta_k\right] \leq \mathbb{E}\left[\theta_k\right] \leq\left(1-\frac{1}{d}\right)^k 2 \varkappa^2 .
$$

Finally, we have

$$
\mathbb{E}\left[\frac{\lambda_{k+1}}{\lambda_k}\right]=\mathbb{E}\left[\frac{\rho_{k+1}}{\rho_k}\right] \stackrel{(57)}{\leq} \mathbb{E}\left[d \varkappa^2 \theta_k\right] \stackrel{(63)}{\leq} \mathbb{E}\left[\left(1-\frac{1}{d}\right)^k 2 d \varkappa^4\right] .
$$
(64)

### D.9 The Proof of Corollary 3.18

*Proof.* We split the proofs of three algorithms into different subsections.

#### D.9.1 Random Broyden Family and BFGS Method

We consider the random variable $X_k=\sigma_k$ or $X_k=\lambda_{k+1} / \lambda_k$ for all $k \geq 0$ in the following derivation.

**Broyden Family Method** The proof is modified from the proof of Theorem 4.7 in Lin et al. [24]. Recall the Section D.8, we obtained the following results for Broyden update

$$
\mathbb{E}\left[X_k\right] \leq\left(1-\frac{1}{d \varkappa^2}\right)^k 2 d \varkappa^2 .
$$
(65)

Note that $\mathbf{X}_k \geq 0$, using Markov's inequality, we have for any $\epsilon>0$,

$$
\mathbb{P}\left(X_k \geq \frac{2 d \varkappa^2}{\epsilon}\left(1-\frac{1}{d \varkappa}\right)^k\right) \leq \frac{\mathbb{E}\left[X_k\right]}{\frac{2 d \varkappa^2}{\epsilon}\left(1-\frac{1}{d \varkappa^2}\right)^k} \stackrel{(65)}{\leq} \epsilon .
$$
(66)

Choosing $\epsilon_k=\delta(1-q) q^k$ for some positive $q<1$, then we have

$$
\begin{aligned}
\mathbb{P}\left(X_k \geq \frac{2 d \varkappa^2}{\epsilon_k}\left(1-\frac{1}{d \varkappa^2}\right)^k, \exists k \in \mathbb{N}\right) & \leq \sum_{k=0}^{\infty} \mathbb{P}\left(X_k \geq \frac{2 d \varkappa^2}{\epsilon_k}\left(1-\frac{1}{d \varkappa^2}\right)^k\right) \\
& \stackrel{(66)}{\leq} \sum_{k=0}^{\infty} \epsilon_k=\sum_{k=0}^{\infty} \delta(1-q) q^k=\delta .
\end{aligned}
$$

Therefore, we obtain with probability $1-\delta$,

$$
X_k \leq\left(\frac{1-\frac{1}{d \varkappa^2}}{q}\right)^k \cdot \frac{2 d \varkappa^2}{(1-q) \delta}, \quad \text { for all } k \in \mathbb{N} .
$$

If we set $q = 1 - \frac{1}{d^2\varkappa^4}$, we could obtain with probability $1 - \delta$, for all $k \in \mathbb{N}$,

$$X_k \leq \frac{2d^3\varkappa^6}{\delta}\left(1 + \frac{1}{d\varkappa^2}\right)^{-k} = \frac{2d^3\varkappa^6}{\delta}\left(1 - \frac{1}{d\varkappa^2+1}\right)^k.$$

Furthermore, it holds with probability $1 - \delta$ that

$$\frac{\lambda_{k+1}}{\lambda_k} \leq \frac{4d^3\varkappa^6}{\delta}\left(1 - \frac{1}{d\varkappa^2+1}\right)^k, \text{ for all } k \in \mathbb{N}, \tag{67}$$

and

$$\sigma_k \leq \frac{4d^3\varkappa^6}{\delta}\left(1 - \frac{1}{d\varkappa^2+1}\right)^k, \text{ for all } k \in \mathbb{N}.$$

Telescoping from $k$ to $0$ in Eq. (67), we get

$$\lambda_k = \lambda_0 \cdot \prod_{i=1}^{k}\frac{\lambda_i}{\lambda_{i-1}} \overset{(67)}{\leq} \lambda_0 \cdot \left(\frac{4d^3\varkappa^6}{\delta}\right)^k \prod_{i=1}^{k}\left(1 - \frac{1}{d\varkappa^2+1}\right)^{i-1}$$
$$= \left(\frac{4d^3\varkappa^6}{\delta}\right)^k\left(1 - \frac{1}{d\varkappa^2+1}\right)^{k(k-1)/2}\lambda_0.$$

Now we combine this result with Theorem 3.15,we give the entire period convergence estimator. Denote by $k_1 \geq 0$ the number of the first iteratiod, for which

$$\left(1 - \frac{1}{4\varkappa^2}\right)^{k_1} \leq \frac{1}{2d+1}.$$

Clearly, $k_1 \leq 4\varkappa^2\ln(2d+1)$. Since the initial point $\mathbf{z}_0$ is close to the saddle point: $M\lambda_0/\mu \leq \ln 2/(8\varkappa^2)$, Combining with Theorem 3.15 by choosing $b = 2$, we have

$$\frac{M\lambda_{k_1}}{\mu} \leq M\left(1 - \frac{1}{4\varkappa^2}\right)^{k_1}\frac{\lambda_0}{\mu} \leq \frac{\ln 2}{8(2d+1)\varkappa^2}, \tag{68}$$

and thus satisfies the initial condition for the Broyden family and the BFGS method in Theorem 3.17. In view of (67) denote $k_2 \geq 0$ the number of the first iteratiod, for which

$$\frac{4d^3\varkappa^6}{\delta}\left(1 - \frac{1}{d\varkappa^2+1}\right)^{k_2} \leq \frac{1}{2}.$$

Clearly, $k_2 \leq (d\varkappa^2 + 1)\ln(8d^3\varkappa^6/\delta)$.

Thus for all $k \geq 0$, we have

$$\lambda_{k_1+k_2+k+1} \overset{(67)}{\leq} \frac{4d^3\varkappa^6}{\delta}\left(1 - \frac{1}{d\varkappa^2+1}\right)^{k_2+k}\lambda_{k_1+k_2+k} \leq \frac{1}{2}\left(1 - \frac{1}{d\varkappa^2+1}\right)^k\lambda_{k_1+k_2+k}.$$

Therefore,

$$\lambda_{k_1+k_2+k} \leq \left(1 - \frac{1}{d\varkappa^2+1}\right)^{k(k-1)/2}\left(\frac{1}{2}\right)^k\lambda_{k_1+k_2},$$

and by Theorem 3.15 we have

$$\lambda_{k_1+k_2} \leq \left(1 - \frac{1}{4\varkappa^2}\right)^{k_1+k_2}\lambda_0.$$

Finally, choose $k_0 = k_1 + k_2 = \mathcal{O}\left(d\varkappa^2\ln(d\varkappa/\delta)\right)$, we obtain

$$\lambda_{k_0+k} \leq \left(1 - \frac{1}{d\varkappa^2+1}\right)^{k(k-1)/2} \cdot \left(\frac{1}{2}\right)^k \cdot \left(1 - \frac{1}{4\varkappa^2}\right)^{k_0}\lambda_0.$$

**BFGS Method**  Similar to the analysis for the random Broyden family method, we obtain with probability $1 - \delta$,

$$X_k \le \left( \frac{1 - \frac{1}{d}}{q} \right)^k \cdot \frac{2d\varkappa^2}{(1-q)\delta}$$

for all $k \in \mathbb{N}$.

If we set $q = 1 - 1/d^2$, we could obtain with probability $1 - \delta$,

$$\lambda_{k+1} \le \frac{4d^3 \varkappa^2}{\delta} \left( 1 - \frac{1}{d+1} \right)^k \lambda_k, \text{ for all } k \in \mathbb{N}, \tag{69}$$

and

$$\sigma_{k+1} \le \frac{4d^3 \varkappa^2}{\delta} \left( 1 - \frac{1}{d+1} \right)^k \sigma_k, \text{ for all } k \in \mathbb{N}.$$

Similar to the above proof, we denote $k_1 \ge 0$ as the number of the first iteration satisfies

$$\left( 1 - \frac{1}{4\varkappa^2} \right)^{k_1} \le \frac{1}{2d+1}. \tag{70}$$

And we denote $k_2 \ge 0$ as the number of the first iteration satisfies

$$\frac{4d^3 \varkappa^2}{\delta} \left( 1 - \frac{1}{d+1} \right)^{k_2} \le \frac{1}{2}. \tag{71}$$

Clearly, $k_1 \le 4\varkappa^2 \ln(2d+1)$ and $k_2 \le (d+1)\ln(8d^3\varkappa^2/\delta)$. The remainder of the proof can follow the analysis in random Broyden methdos. We only need to replace all the term of $\left( 1 - \frac{1}{d\varkappa^2+1} \right)$ to $\left( 1 - \frac{1}{d+1} \right)$. The reason is that (69) provides a faster convergence result for random BFGS update, rather than (67). Set $k_0 = k_1 + k_2 = \mathcal{O}(\max\{d, \varkappa^2\} \ln(d\varkappa/\delta))$, we obtain

$$\lambda_{k_0+k} \le \left( 1 - \frac{1}{d+1} \right)^{k(k-1)/2} \cdot \left( \frac{1}{2} \right)^k \cdot \left( 1 - \frac{1}{4\varkappa^2} \right)^{k_0} \lambda_0.$$

### D.9.2   Random SR1 Method

**SR1 Method**  We denote $X_k = \lambda_{k+1}/\lambda_k$ or $X_k = 2d\varkappa^2\theta_k$ for all $k \ge 0$ in the following derivation.

According to the results of (63), (64) and similar to the analysis for Broyden family method, we obtain with probability $1 - \delta$ for SR1 update that

$$X_k \le \left( \frac{1 - \frac{1}{d}}{q} \right)^k \cdot \frac{2d\varkappa^4}{(1-q)\delta}$$

for all $k \in \mathbb{N}$.

If we set $q = 1 - 1/d^2$, we have

$$\lambda_{k+1} \le \frac{4d^3 \varkappa^4}{\delta} \left( 1 - \frac{1}{d+1} \right)^k \lambda_k$$

for all $k \in \mathbb{N}$. Similar to the above proof, we denote $k_1 \ge 0$ as the number of the first iteration satisfies

$$\left( 1 - \frac{1}{4\varkappa^2} \right)^{k_1} \le \frac{1}{(2d\varkappa^2 + 1)}.$$

Since the initial point $\mathbf{z}_0$ is close to the saddle point: $M\lambda_0/\mu \le \ln 2/(8\varkappa^2)$, Combining with Theorem 3.15 by choosing $b = 2$, we have

$$\frac{M\lambda_{k_1}}{\mu} \le M \left( 1 - \frac{1}{4\varkappa^2} \right)^{k_1} \frac{\lambda_0}{\mu} \le \frac{\ln 2}{8(2d\varkappa^2 + 1)\varkappa^2},$$

which satisfies the initial condition for SR1 method in Theorem 3.17. And we denote $k_2 \geq 0$ as the number of the first iteration satisfies

$$\frac{4d^3 \varkappa^4}{\delta} \left(1 - \frac{1}{d+1}\right)^{k_2} \leq \frac{1}{2}.$$

We have $k_1 \leq 4\varkappa^2 \ln(2d\varkappa^2 + 1)$ and $k_2 \leq (d+1)\ln(8d^3\varkappa^4/\delta)$.

Similar to above analysis, we obtain

$$\lambda_{k_0+k} \leq \left(1 - \frac{1}{d+1}\right)^{k(k-1)/2} \cdot \left(\frac{1}{2}\right)^k \cdot \left(1 - \frac{1}{4\varkappa^2}\right)^{k_0} \lambda_0$$

by setting $k_0 = k_1 + k_2 = \mathcal{O}(\max\{d, \varkappa^2\} \ln(d\varkappa/\delta))$.

$\square$

# E   Experimental Details

We provide some details for our experiments in this section. Our experiments are conducted on a work station with 56 Intel(R) Xeon(R) Gold 6132 CPU @ 2.60GHz and 256GB memory. We use MATLAB 2021a to run the code and the operating system is Ubuntu 20.04.2.

## E.1   AUC Maximization

The gradient of the object function at $\mathbf{z} = [\mathbf{x}; \mathbf{y}] = [\mathbf{w}; u; v; y]$ is

$$\mathbf{g}(\mathbf{z}) = \nabla f(\mathbf{z}) = \frac{1}{n} \sum_{i=1}^{n} \begin{bmatrix} \nabla_{\mathbf{w}} f_i(\mathbf{z}) \\ \nabla_u f_i(\mathbf{z}) \\ \nabla_v f_i(\mathbf{z}) \\ \nabla_y f_i(\mathbf{z}) \end{bmatrix},$$

where

$$\nabla_{\mathbf{w}} f_i(\mathbf{z}) = \lambda \mathbf{w} + 2(1-p)(\mathbf{w}^\top \mathbf{a}_i - u - 1 - y))\mathbf{a}_i \mathbb{I}_{b_i=1} + 2p(\mathbf{w}_i^\top \mathbf{a}_i - v + 1 + y)\mathbf{a}_i \mathbb{I}_{b_i=-1},$$
$$\nabla_u f_i(\mathbf{z}) = \lambda u - 2(1-p)(\mathbf{w}^\top \mathbf{a}_i - u)\mathbb{I}_{b_i=1},$$
$$\nabla_v f_i(\mathbf{z}) = \lambda v - 2p(\mathbf{w}^\top \mathbf{a}_i - v)\mathbb{I}_{b_i=-1},$$
$$\nabla_y f_i(\mathbf{z}) = -2p(1-p)y + 2p\mathbf{w}^\top \mathbf{a}_i \mathbb{I}_{b_i=-1} - 2(1-p)\mathbf{w}^\top \mathbf{a}_i \mathbb{I}_{b_i=1}.$$

The Hessian-vector of the object function is

$$\nabla^2 f(\mathbf{z})\mathbf{h} = \hat{\mathbf{H}}(\mathbf{z})\mathbf{h} = \frac{1}{n} \sum_{i=1}^{n} \begin{bmatrix} (\nabla^2 f_i(\mathbf{z})\mathbf{h})_{\mathbf{w}} \\ (\nabla^2 f_i(\mathbf{z})\mathbf{h})_u \\ (\nabla^2 f_i(\mathbf{z})\mathbf{h})_v \\ (\nabla^2 f_i(\mathbf{z})\mathbf{h})_y \end{bmatrix},$$

where $\mathbf{h} = [\mathbf{h}_{\mathbf{w}}; \mathbf{h}_u; \mathbf{h}_v; \mathbf{h}_y]$ such that

$$(\nabla^2 f_i(\mathbf{z})\mathbf{h})_{\mathbf{w}} = \lambda \mathbf{h}_{\mathbf{w}} + 2(1-p)(\langle \mathbf{a}_i, \mathbf{h}_{\mathbf{w}} \rangle - \mathbf{h}_u - \mathbf{h}_y)\mathbf{a}_i \mathbb{I}_{b_i=1}$$
$$+ 2p(\langle \mathbf{a}_i, \mathbf{h}_{\mathbf{w}} \rangle - \mathbf{h}_v + \mathbf{h}_y)\mathbf{a}_i \mathbb{I}_{b_i=-1},$$
$$(\nabla^2 f_i(\mathbf{z})\mathbf{h})_u = -2(1-p)\mathbf{a}_i^\top \mathbf{h}_{\mathbf{w}} \mathbb{I}_{b_i=1} + (\lambda + 2(1-p)\mathbb{I}_{b_i=1})\mathbf{h}_u,$$
$$(\nabla^2 f_i(\mathbf{z})\mathbf{h})_v = -2p\mathbf{a}_i^\top \mathbf{h}_{\mathbf{w}} \mathbb{I}_{b_i=-1} + (\lambda + 2p\mathbb{I}_{b_i=-1})\mathbf{h}_v,$$
$$(\nabla^2 f_i(\mathbf{z})\mathbf{h})_y = -2p(1-p)y + 2p\mathbf{a}_i^\top \mathbf{h}_{\mathbf{w}} \mathbb{I}_{b_i=-1} - 2(1-p)\mathbf{a}_i^\top \mathbf{h}_{\mathbf{w}} \mathbb{I}_{b_i=1}.$$

Note that the Hessian-vector can be achieved in $\mathcal{O}(nd)$ flops and it guarantees $\mathcal{O}(nd + d^2)$ complexity for each iteration.

For baseline method extragradient (Algorithm 8), we tune the stepsize from $\{0.01, 0.05, 0.1, 0.5\}$. For RaBFGSv1-Q (Algorithm 2), we let $\mathbf{G}_0 = 3\mathbf{I}$ for "a9a", "w8a" and $\mathbf{G}_0 = 30\mathbf{I}$ for "sido0". For RaBFGSv2-Q (Algorithm 3), we let $\mathbf{G}_0 = 3\mathbf{I}$ for "a9a" and "w8a". We do not run RaBFGSv2-Q on

---
**Algorithm 8** Extragradient Method
---
1: **Input:** $\mathbf{z}_0 \in \mathbb{R}^n$, and $\eta > 0$.

2: **for** $k = 0, 1, \ldots$

3:    $\mathbf{x}_{k+1/2} = \mathbf{x}_k - \eta \nabla_{\mathbf{x}} f(\mathbf{x}_k, \mathbf{y}_k)$

4:    $\mathbf{y}_{k+1/2} = \mathbf{y}_k + \eta \nabla_{\mathbf{y}} f(\mathbf{x}_k, \mathbf{y}_k)$

5:    $\mathbf{x}_{k+1} = \mathbf{x}_k - \eta \nabla_{\mathbf{x}} f(\mathbf{x}_{k+1/2}, \mathbf{x}_{k+1/2})$

6:    $\mathbf{y}_{k+1} = \mathbf{y}_k + \eta \nabla_{\mathbf{y}} f(\mathbf{x}_{k+1/2}, \mathbf{x}_{k+1/2})$

7: **end for**
---

"sido0" because this algorithm is not efficient for high-dimensional problem as we have mentioned in Remark 3.5. For RaSR1-Q (Algorithm 4), we let $\mathbf{G}_0 = 5\mathbf{I}$ for "a9a", "w8a" and $\mathbf{G}_0 = 30\mathbf{I}$ for "sido0".

The dataset "sido0" comes from Causality Workbench [17] and the other datasets can be downloaded from LIBSVM repository [8].

### E.2   Adversarial Debiasing

The minimax formulation can be rewritten as

$$\min_{\mathbf{x} \in \mathbb{R}^m} \max_{y \in \mathbb{R}} \frac{1}{n} \sum_{i=1}^{n} f_i(\mathbf{x}, y; \mathbf{a}_i, b_i, c_i, \lambda, \gamma, \beta)$$

where $f_i$ is defined as
$$f_i(\mathbf{x}, y; \mathbf{a}_i, b_i, c_i, \lambda, \gamma, \beta) = \log(1 + \exp(-b_i \mathbf{a}_i^\top \mathbf{x})) - \beta \log(1 + \exp(-c_i \mathbf{a}_i^\top \mathbf{x} y)) + \lambda \|\mathbf{x}\|^2 - \gamma y^2.$$
We define

$$p_i = \frac{1}{1 + \exp(b_i \mathbf{a}_i^\top \mathbf{x})} \qquad \text{and} \qquad q_i = \frac{1}{1 + \exp(c_i y \mathbf{a}_i^\top \mathbf{x})}.$$

Then the gradient of the object function at $\mathbf{z} = [\mathbf{x}; y]$ is

$$\mathbf{g}(\mathbf{z}) = \nabla f(\mathbf{z}) = \frac{1}{n} \sum_{i=1}^{n} \begin{bmatrix} \nabla_{\mathbf{x}} f_i(\mathbf{z}) \\ \nabla_y f_i(\mathbf{z}) \end{bmatrix},$$

where

$$\nabla_{\mathbf{x}} f_i(\mathbf{z}) = -p_i b_i \mathbf{a}_i + \beta q_i c_i y \mathbf{a}_i + 2\lambda \mathbf{x} \qquad \text{and} \qquad \nabla_y f_i(\mathbf{z}) = \beta c_i \mathbf{a}_i^\top x q_i - 2\gamma y.$$

The Hessian-vector of the object function is

$$\nabla^2 f(\mathbf{z})\mathbf{h} = \hat{\mathbf{H}}(\mathbf{z})\mathbf{h} = \frac{1}{n} \sum_{i=1}^{n} \begin{bmatrix} (\nabla^2 f_i(\mathbf{z})\mathbf{h})_{\mathbf{x}} \\ (\nabla^2 f_i(\mathbf{z})\mathbf{h})_y \end{bmatrix}.$$

where $\mathbf{h} = [\mathbf{h}_{\mathbf{x}}; \mathbf{h}_y]$ such that

$$(\nabla^2 f_i(\mathbf{z})\mathbf{h})_{\mathbf{x}} = (p_i(1 - p_i)\mathbf{a}_i^\top \mathbf{h}_{\mathbf{x}} - q_i(1 - q_i)\beta y^2 \mathbf{a}_i^\top \mathbf{h}_{\mathbf{x}})\mathbf{a}_i + 2\lambda \mathbf{h}_{\mathbf{x}}$$
$$- (q_i(1 - q_i)\beta y \mathbf{a}_i^\top \mathbf{x} - q_i \beta c)\mathbf{h}_y \mathbf{a}_i,$$
$$(\nabla^2 f_i(\mathbf{z})\mathbf{h})_y = -\beta y q_i(1 - q_i)\mathbf{a}_i^\top \mathbf{x} \mathbf{a}_i^\top \mathbf{h}_{\mathbf{x}} + q_i \beta c \mathbf{a}_i^\top \mathbf{h}_{\mathbf{x}} - q_i(1 - q_i)\beta (\mathbf{a}_i^\top \mathbf{x})^2 \mathbf{h}_y - 2\gamma \mathbf{h}_y.$$

The Hessian-vector can be achieved in $\mathcal{O}(nd)$ flops and it guarantees $\mathcal{O}(nd + d^2)$ complexity for each iteration.

**Data Preparation**   The experiments are based on the datasets of fairness aware machine learning [35]. Following the reprocessing of Chang and Lin [8], Platt [32], we convert the features of the original datasets into binary for our experiments. Concretely, the continuous features are discretized into quantiles, and each quantile is represented by a binary feature. Also, a categorical feature with $C$ categories is converted to $C$ binary features. More specifically, for the "adults" dataset, we transform the 13 features of it into 122 binary features and choose the feature of "gender" as the protected feature. For the "law school" dataset, we transform the 11 features of it into 379 binary features and choose the feature of "gender" as the protected feature. For the "bank marketing" dataset, we transform 16 features of it into 3879 binary features and choose "marital" as the protected feature.

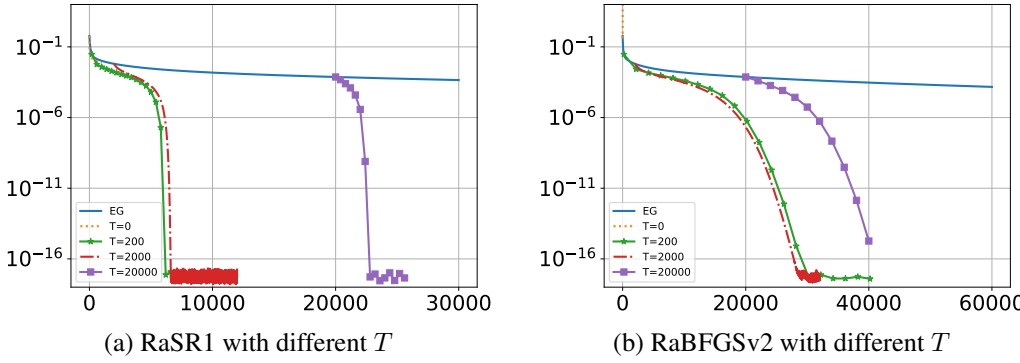

| (a) RaSR1 with different $T$ | (b) RaBFGSv2 with different $T$ |

Figure 3: We demonstrate iteration numbers vs. $\|\mathbf{g}(\mathbf{z})\|_2$ for adversarial debiasing model on datasets "law school" ($d = 380$, $n = 20427$).

**Assumptions Validation**   We can verify the objective function satisfies our assumptions. Since the convergence results in Section 3.3 are local, we only need to show that $f$ satisfies Assumption 2.1 and 2.2 in a local region around $\mathbf{z}^*$. Hence, we only needs to consider $(\mathbf{x}, y)$ such that $\|\mathbf{x}\| \leq D_1$ and $\|y\| \leq D_2$ for some $D_1 > 0$ and $D_2 > 0$.

Using the notation in Appendix E.2, we have

$$\left(2\lambda - \left\|\frac{\beta y^2}{4n}\sum_{i=1}^{n}\mathbf{a}_i^\top\mathbf{a}_i\right\|\right)\mathbf{I} \preceq \nabla_{\mathbf{xx}}^2 f(\mathbf{z}) \preceq \left(\|\frac{1}{4n}\sum_{i=1}^{n}\mathbf{a}_i^\top\mathbf{a}_i\| + 2\lambda\right)\mathbf{I},$$

$$2\gamma \leq -\nabla_{yy}^2 f(\mathbf{z}) \leq \left(\frac{1}{4n}\sum_{i=1}^{n}\left(\beta\mathbf{a}_i^\top\mathbf{x}\right)^2 + 2\gamma\right).$$

Additionally, the feature $\{\mathbf{a}_i\}$ is sparse due to the transform, which means the upper bound of $\left\|\sum_{i=1}^{n}\mathbf{a}_i^\top\mathbf{a}_i\right\|$ is small. We can guarantee $2\lambda - \left\|\frac{\beta D_2^2}{4n}\sum_{i=1}^{n}\mathbf{a}_i^\top\mathbf{a}_i\right\| > 0$ by choosing $\lambda$ properly. Theoretically, setting $\mu = \min\left\{2\lambda - \left\|\frac{\beta D_2^2}{4n}\sum_{i=1}^{n}\mathbf{a}_i^\top\mathbf{a}_i\right\|, 2\gamma\right\}$ can guarantee $f$ satisfies Assumption 2.2.

**Experiment Parameters**   For experiments, we set $\mathbf{G}_0 = 2\mathbf{I}$ for the proposed quasi-Newton methods. For RaBFGSv1-G (Algorithm 5), we set $M = 0$ for "adults", $M = 1$ for "law school" and "bank market". For RaBFGSv2-G (Algorithm 6), we set $M = 1$ for "adults" and "law school". For RaSR1-G (Algorithm 7), we set $M = 1$ for all three datasets.

We tune the stepsize of EG from $\{0.01, 0.05, 0.1, 0.5\}$ and run it with 4000, 20000 and 40000 iterations for the "adults", "law school" and "Bank market" respectively as warm up to obtain $\mathbf{z}_0$ as initial point. Then we evaluate all algorithms (including the baseline algorithm EG) by starting with $\mathbf{z}_0$ and achieve the result shown in Figure 2. Since each algorithm has the identical behavior in the warm up stage, we only present the curves of iterations vs. $\|\mathbf{g}(\mathbf{z})\|_2$ and CPU time vs. $\|\mathbf{g}(\mathbf{z})\|_2$ after warm up stage in Figure 2.

We run additional experiments on RaSR1 and RaBFGSv2 for "law school" with different number rounds of extragradient iteration as warm-up. The number of iterations $T$ is selected from $\{0, 200, 2000, 20000\}$. The results in Figure 3 show that our methods could converge even for small $T$.

### E.3   Two Stages Convergence Behavior

For most of cases, our algorithms enter the local region of superlinear convergence quickly, so that the two-period convergence is not very clear on the figure. Let's have a look at RaBFGSv2-Q in (b) of Figure 1 (the purple curve). We can observe two-period convergence behavior clearly. The first period (linear convergence) roughly corresponds to the first 2000 rounds of iterations on the figure and the second period (superlinear convergence) roughly corresponds to the later iterations. The

convergence behavior of first period for RaBFGSv2-Q looks worse than the first-order method (EG), which is reasonable since the linear convergence rate of RaBFGSv2-Q in the first period depends on $\kappa^2$ while the linear convergence rate of EG depends on $\kappa$.

# F   Extension for Solving Nonlinear Equations

In this section, we extend our algorithms for solving general nonlinear equations. Concretely, we consider finding the solution of the system

$$\mathbf{g}(\mathbf{z}) = \mathbf{0}. \tag{72}$$

The remainders of this paper do not require the operator $\mathbf{g}(\cdot)$ related to some minimax problem and it could be any differentiable function from $\mathbb{R}^d$ to $\mathbb{R}^d$. We use $\hat{\mathbf{H}}(\mathbf{z})$ to present the Jacobian of $\mathbf{g}(\cdot)$ at $\mathbf{z} \in \mathbb{R}^d$ and still follow the notation $\mathbf{H}(\mathbf{z}) \overset{\text{def}}{=} \left(\hat{\mathbf{H}}(\mathbf{z})\right)^2$. We suppose the nonlinear equation satisfies the following conditions.

**Assumption F.1.** The function $\mathbf{g} : \mathbb{R}^d \to \mathbb{R}^d$ is differentiable and its Jacobian $\hat{\mathbf{H}} : \mathbb{R}^d \to \mathbb{R}^{d \times d}$ is $L_2$-Lipschitz continuous. That is, for all $\mathbf{z}, \mathbf{z}' \in \mathbb{R}^d$, we have

$$\left\| \hat{\mathbf{H}}(\mathbf{z}) - \hat{\mathbf{H}}(\mathbf{z}') \right\| \le L_2 \|\mathbf{z} - \mathbf{z}'\|. \tag{73}$$

**Assumption F.2.** There exists a solution $\mathbf{z}^*$ of equation (72) such that $\hat{\mathbf{H}}(\mathbf{z}^*)$ is non-singular. Additionally, we assume the smallest and largest singular values of $\hat{\mathbf{H}}(\mathbf{z}^*)$ are $\mu$ and $L$ respectively.

We still denote the condition number as $\varkappa \overset{\text{def}}{=} L/\mu$. Note that the saddle point problem (1) under Assumption 2.1 and 2.2 is a special case of solving nonlinear equation (72) under Assumption F.1 and F.2.

Similar to previous section, the design of the algorithms is based on approximating the auxiliary matrix $\mathbf{H}(\mathbf{z})$. Hence, we start from considering its smoothness.

**Lemma F.3.** *Under Assumption F.1 and F.2, we have*

$$\|\mathbf{H}(\mathbf{z}) - \mathbf{H}(\mathbf{z}')\| \le 4L_2 L \|\mathbf{z} - \mathbf{z}'\|, \tag{74}$$

*and*

$$\frac{\mu^2}{2} \mathbf{I} \preceq \mathbf{H}(\mathbf{z}) \preceq 4L^2 \mathbf{I}, \tag{75}$$

*for all* $\mathbf{z}, \mathbf{z}' \in \left\{ \mathbf{z} : \|\mathbf{z} - \mathbf{z}^*\| \le \frac{L}{8\varkappa^2 L_2} \right\}$.

*Proof.* We define the local neighbor of the solution $\mathbf{z}^*$ as follows

$$\Omega^* \overset{\text{def}}{=} \{\mathbf{z} : \|\mathbf{z} - \mathbf{z}^*\| \le D\}, \quad \text{where } D = \frac{\mu^2}{8L_2 L}.$$

Assumption F.2 implies $\sigma_{\min}\left(\mathbf{H}(\mathbf{z}^*)\right) = \mu^2$ and $\sigma_{\max}\left(\mathbf{H}(\mathbf{z}^*)\right) = L^2$.

Since $\mathbf{H}(\mathbf{z}^*)$ is invertible and symmetric, we have $\mathbf{H}(\mathbf{z}^*) \succ 0$ and we have restricted $\mathbf{z}, \mathbf{z}' \in \Omega^*$, it holds that

$$\left\| \hat{\mathbf{H}}(\mathbf{z}) \right\| \le \left\| \hat{\mathbf{H}}(\mathbf{z}) - \hat{\mathbf{H}}(\mathbf{z}^*) \right\| + \left\| \hat{\mathbf{H}}(\mathbf{z}^*) \right\| \le L_2 \|\mathbf{z} - \mathbf{z}^*\| + \left\| \hat{\mathbf{H}}(\mathbf{z}^*) \right\| \le L_2 D + L, \tag{76}$$

which implies

$$\mathbf{H}(\mathbf{z}) \preceq (L_2 D + L)^2 \mathbf{I} \tag{77}$$

for all $\mathbf{z} \in \Omega^*$. According to (77), we have

$$
\begin{aligned}
\|\mathbf{H}(\mathbf{z}) - \mathbf{H}(\mathbf{z}')\| &= \left\| \hat{\mathbf{H}}(\mathbf{z})^2 - \hat{\mathbf{H}}(\mathbf{z}')^2 \right\| \\
&\le \left\| \hat{\mathbf{H}}(\mathbf{z}) \left( \hat{\mathbf{H}}(\mathbf{z}) - \hat{\mathbf{H}}(\mathbf{z}') \right) \right\| + \left\| \left( \hat{\mathbf{H}}(\mathbf{z}) - \hat{\mathbf{H}}(\mathbf{z}') \right) \hat{\mathbf{H}}(\mathbf{z}') \right\| \\
&\le L_2 \left( \left\| \hat{\mathbf{H}}(\mathbf{z}) \right\| + \left\| \hat{\mathbf{H}}(\mathbf{z}') \right\| \right) \|\mathbf{z} - \mathbf{z}'\|
\end{aligned}
$$

$$\overset{(76)}{\leq} \quad 2L_2(L_2D + L)\|\mathbf{z} - \mathbf{z}'\|.$$

Combining above result with the Weyl's inequality for singular values [19, Theorem 3.3.16], we have

$$\sigma_{\min}(\mathbf{H}(\mathbf{z}^*)) - 2L_2(L_2D + L)\|\mathbf{z} - \mathbf{z}^*\| \leq \sigma_{\min}(\mathbf{H}(\mathbf{z})).$$

Since the definition of $D$ indicates $2L_2(L_2D + L)D \leq \frac{\mu^2}{2}$ and $L_2D \leq L$, we have

$$\frac{\mu^2}{2}\mathbf{I} \preceq \mathbf{H}(\mathbf{z}) \preceq 4L^2\mathbf{I},$$

and

$$\|\mathbf{H}(\mathbf{z}) - \mathbf{H}(\mathbf{z}')\| \leq 4L_2L\|\mathbf{z} - \mathbf{z}'\|$$

$\square$

Then we have the property similar to strongly self-concordance like Lemma 3.11.

**Lemma F.4.** *For all* $\mathbf{z}, \mathbf{z}', \mathbf{w} \in \left\{ \mathbf{z} : \|\mathbf{z} - \mathbf{z}^*\| \leq \frac{L}{8\varkappa^2 L_2} \right\}$, *we have:*

$$\mathbf{H}(\mathbf{z}) - \mathbf{H}(\mathbf{z}') \preceq M\|\mathbf{z} - \mathbf{z}'\|\mathbf{H}(\mathbf{w}) \tag{78}$$

*with* $M = 8\varkappa^2 L_2/L$

*Proof.* According to Lemma F.3, we have

$$
\begin{aligned}
\mathbf{H}(\mathbf{z}) - \mathbf{H}(\mathbf{z}') \quad &\overset{(74)}{\preceq} \quad 4L_2L\|\mathbf{z} - \mathbf{z}'\|\mathbf{I} \\
&\overset{(75)}{\preceq} \quad \frac{8L_2L}{\mu^2}\|\mathbf{z} - \mathbf{z}'\|\mathbf{H}(\mathbf{w}) \\
&= \quad \frac{8\varkappa^2 L_2}{L}\|\mathbf{z} - \mathbf{z}'\|\mathbf{H}(\mathbf{w}).
\end{aligned}
$$

for all $\mathbf{z}, \mathbf{z}' \in \Omega^*$.

$\square$

After above preparation, we can directly apply Algorithm 5, 6 and 7 with $M = 8\varkappa^2 L_2/L$ and $\mathbf{G}_0 = 4L^2\mathbf{I}$ to find the solution of (72). Our convergence analysis is still based on the measure of $\lambda_k \overset{\text{def}}{=} \|\nabla(\mathbf{z}_k)\|$. Different from the setting of saddle point problems, the properties shown in Lemma F.3 and F.4 only hold locally. Hence, we introduce the following lemma to show $\mathbf{z}_k$ generated from the algorithms always lies in the neighbor of solution $\mathbf{z}^*$.

**Lemma F.5.** *Solving general nonlinear equations* (72) *under Assumption F.1 and F.2 by proposed greedy and random quasi-Newton methods (Algorithm 5, 6 and 7) with* $M = 8\varkappa^2 L_2/L$ *and* $\mathbf{G}_0 = 4L^2\mathbf{I}$, *if the initial point* $\mathbf{z}_0$ *is sufficiently close to the solution* $\mathbf{z}^*$ *such that*

$$\frac{M\lambda_0}{\mu} \leq \frac{\ln 2}{64\sqrt{2}\varkappa^2}, \tag{79}$$

*then for all* $k \geq 0$, *we have*

$$\lambda_k \leq \left(1 - \frac{1}{32\varkappa^2}\right)^k \lambda_0, \tag{80}$$

*which means* $\mathbf{z}_k \in \left\{ \mathbf{z} : \|\mathbf{z} - \mathbf{z}^*\| \leq \frac{L}{8\varkappa^2 L_2} \right\}$.

*Proof.* We prove this lemma by induction. For $k = 0$, it is obviously. Suppose the statement holds for all $k' \leq k$. Then for all $k' = 0, \ldots, k$, we have $\mathbf{z}_{k'} \in \Omega^*$ and $\mathbf{z}_{k'}$ holds that (75), (74) and (78). By Theorem 3.15, we guarantee

$$\mathbf{H}_{k'} \leq \mathbf{G}_{k'} \leq 16\varkappa^2\mathbf{H}_{k'}. \tag{81}$$

For $k' = k + 1$, according to the proof of Lemma 3.14, we have

$$\mathbf{g}_{k+1} = \underbrace{(\mathbf{I} - \hat{\mathbf{H}}_k\mathbf{G}_k^{-1}\hat{\mathbf{H}}_k)\mathbf{g}_k}_{\mathbf{a}_k} + \underbrace{\int_0^1 \left[\hat{\mathbf{H}}(\mathbf{z}_k + s(\mathbf{z}_{k+1} - \mathbf{z}_k)) - \hat{\mathbf{H}}(\mathbf{z}_k)\right](\mathbf{z}_{k+1} - \mathbf{z}_k)ds}_{\mathbf{b}_k}.$$

Using Lemma B.2 and the result of (81), we have

$$\|\mathbf{I} - \hat{\mathbf{H}}_k \mathbf{G}_k^{-1} \hat{\mathbf{H}}_k\| \le 1 - \frac{1}{16\varkappa^2}$$

and

$$\|\mathbf{b}_k\| \le \frac{L_2}{\mu^2} \lambda_k^2.$$

Combing above results, we have

$$\lambda_{k+1} \le \left(1 - \frac{1}{16\varkappa^2}\right) \lambda_k + \frac{L_2}{\mu^2} \lambda_k^2 \le \left(1 - \frac{1}{32\varkappa^2}\right) \lambda_k.$$

Thus we always have $\|\mathbf{z}_{k+1} - \mathbf{z}_k\| \le \frac{1}{\mu} \lambda_{k+1} \le \lambda_0 \le D$. By induction, we finish the proof. $\qquad\square$

Based on Lemma F.5, we establish the following theorem to show the algorithms also have local superlinear convergence for solving nonlinear equations.

**Theorem F.6.** *Solving general nonlinear equations* (72) *under Assumption F.1 and F.2 by proposed quasi-Newton methods (Algorithm 5, 6 and 7) with $M = 8\varkappa^2 L_2/L$ and $\mathbf{G}_0 = 4L^2\mathbf{I}$, if the initial point $\mathbf{z}_0$ is sufficiently close to the solution $\mathbf{z}^*$ such that*

$$\frac{M\lambda_0}{\mu} \le \frac{\ln 2}{64\sqrt{2}\varkappa^2},$$

*with probability $1 - \delta$ for any $\delta \in (0, 1)$, we have the following results.*

1. *For random Broyden family method (Algorithm 5), we have*

$$\lambda_{k_0+k} \le \left(1 - \frac{1}{8d\varkappa^2 + 1}\right)^{\frac{k(k-1)}{2}} \left(\frac{1}{2}\right)^k \left(1 - \frac{1}{32\varkappa^2}\right)^{k_0} \lambda_0$$

   *for all $k > 0$ and $k_0 = \mathcal{O}\left(d\varkappa^2 \ln(d\varkappa/\delta)\right)$.*

2. *For random BFGS/SR1 method (Algorithm 6, 7), we have*

$$\lambda_{k_0+k} \le \left(1 - \frac{1}{d+1}\right)^{\frac{k(k-1)}{2}} \left(\frac{1}{2}\right)^k \left(1 - \frac{1}{32\varkappa^2}\right)^{k_0} \lambda_0$$

   *for all $k > 0$ and $k_0 = \mathcal{O}\left(\max\{d, \varkappa^2\} \ln(d\varkappa/\delta)\right)$.*

*Proof.* We denote

$$L' \stackrel{\text{def}}{=} 2L, \qquad \mu' \stackrel{\text{def}}{=} \sqrt{2}\mu \qquad \text{and} \qquad \varkappa' \stackrel{\text{def}}{=} \frac{L'}{\mu'} = \frac{2\sqrt{2}L}{\mu}.$$

The initial condition on $\lambda_0$ and Lemma F.5 means all the points $\mathbf{z}_k$ generated from our algorithms are located in $\Omega^*$. Thus we can directly use the results of 3.18 by replacing $\varkappa$ and $\mu$ of the Corollary 3.18 into $\varkappa' = \frac{L'}{\mu'} = 2\sqrt{2}\varkappa$ and $\mu' = \frac{1}{\sqrt{2}}\mu$ here. $\qquad\square$

**Discussion** Recently, Lin et al. [23], Ye et al. [46] showed Broyden's methods have explicit local superlinear convergence rate for solving nonlinear equations, we compare our methods with theirs as follow.

- For the assumptions, they suppose $\|\hat{\mathbf{H}}(\mathbf{z}) - \hat{\mathbf{H}}(\mathbf{z}^*)\| \le L_2 \|\mathbf{z} - \mathbf{z}^*\|$ for any $\mathbf{z} \in \mathbb{R}^d$, which is weaker than Assumption F.1 of ours.

- For the convergence rate, we compare our results with Lin et al. [23], Ye et al. [46] in Table 2. The superlinear convergence rate of Lin et al. [23] is $\mathcal{O}((1/\sqrt{k})^k)$, which is worse than ours. The one of Ye et al. [46] is $\mathcal{O}((1 - 1/(d+1))^{k(k-1)/4})$, which is comparable to ours.

Table 2: Comparison of Upper Bound. The measure of Lin et al. [23], Ye et al. [46] is the Euclidean distance $\|\mathbf{z}_k - \mathbf{z}^*\|$. We denote $c = \left\|(\hat{\mathbf{H}}^*)^{-1}\right\|$ and $\sigma_0 = \left\|(\hat{\mathbf{H}}^*)^{-1}(\hat{\mathbf{G}}_0 - \hat{\mathbf{H}}^*)\right\|$. The measure of ours is the gradient norm $\|\mathbf{g}(\mathbf{z}_k)\|$.

| Algorithms | Upper Bound of $\lambda_{k+k_0}/\lambda_0$ |
|---|---|
| [23, Theorem 4.4] | $\left(\dfrac{7(\sigma_0 + \sqrt{cL_2\|\mathbf{z}_0 - \mathbf{z}^*\|})}{\sqrt{k+k_0}}\right)^{k+k_0}$ |
| [46, Theorem 4.5] | $\left(\dfrac{4d^2e}{\delta}\right)^{k+k_0}\left(1 - \dfrac{1}{d+1}\right)^{(k+k_0)(k+k_0-1)/4}$ |
| Theorem F.6 of this paper | $\left(1 - \dfrac{1}{d+1}\right)^{k(k-1)/2}\left(\dfrac{1}{2}\right)^k\left(1 - \dfrac{1}{32\varkappa^2}\right)^{k_0}$ |

Table 3: Comparison of Initial Condition

| Reference | Initial Condition |
|---|---|
| [23, Theorem 4.4] | $48L_2\left\|(\hat{\mathbf{H}}^*)^{-1}\right\|\|\mathbf{z}_0 - \mathbf{z}^*\| + \left\|(\hat{\mathbf{H}}^*)^{-1}(\hat{\mathbf{G}}_0 - \hat{\mathbf{H}}^*)\right\|_F \leq \dfrac{1}{3}$ |
| [46, Theorem 4.3] | $48L_2\left\|(\hat{\mathbf{H}}^*)^{-1}\right\|\|\mathbf{z}_0 - \mathbf{z}^*\| + \left\|(\hat{\mathbf{H}}^*)^{-1}\right\|\|\hat{\mathbf{G}}_0 - \hat{\mathbf{H}}(\mathbf{z}_0)\| \leq \dfrac{1}{3}$ |
| Theorem F.6 of this paper | $\lambda_0 \leq \dfrac{\ln(2)\mu^5}{512\sqrt{2}L^3L_2}$ |
| [37, Theorem 4.7] (min) | $\langle\nabla f(\mathbf{z}_0), \nabla^2 f(\mathbf{z}_0)^{-1}\nabla f(\mathbf{z}_0)\rangle^{1/2} \leq \dfrac{\ln(3/2)\mu^{5/2}}{4L_2L}$ |
| [24, Corollary 21] (min) | $\langle\nabla f(\mathbf{z}_0), \nabla^2 f(\mathbf{z}_0)^{-1}\nabla f(\mathbf{z}_0)\rangle^{1/2} \leq \dfrac{\ln(3/2)\mu^{5/2}}{4L_2L}$ |

- As for the initial condition, we compare of us with Lin et al. [23], Ye et al. [46] and quasi-Newton methods for minimization problem Lin et al. [24], Rodomanov and Nesterov [37] in Table 3. Our algorithms only require $\mathbf{z}_0$ be sufficiently close to the solution, while Lin et al. [23], Ye et al. [46] need stronger initial condition. Concretely, Lin et al. [23] requires $\hat{\mathbf{G}}_0$ be sufficiently close to $\hat{\mathbf{H}}^*$ and Ye et al. [46] requires $\hat{\mathbf{G}}_0$ be sufficiently close to $\hat{\mathbf{H}}(\mathbf{z}_0)$.

In practice, there is no general method to achieve an initial matrix $\mathbf{G}_0$ that is sufficiently closed to the exact Jacobian at $\mathbf{z}_0$ or $\mathbf{z}^*$ and computing the inverse of $\mathbf{G}_0$ always requires huge computation. Another way is using the matrix of the form $\mathbf{G}_0 = L\mathbf{I}$ as initialization, however, this strategy does not always work. we their methods on adversarial debiasing model of Section 5.2.

We tune the stepsize of EG from $\{0.01, 0.05, 0.1, 0.5\}$ and run it with 10000 and 20000 iterations for the "adults" and "law school" respectively as warm up to obtain $\mathbf{z}_0$ as initial point. Start with such initial point $\mathbf{z}_0$, our methods always converge as is shown in Figure 2 while the results in Figure 4 and 5 show that Broyden methods of Lin et al. [23] and Ye et al. [46] fail to converge.

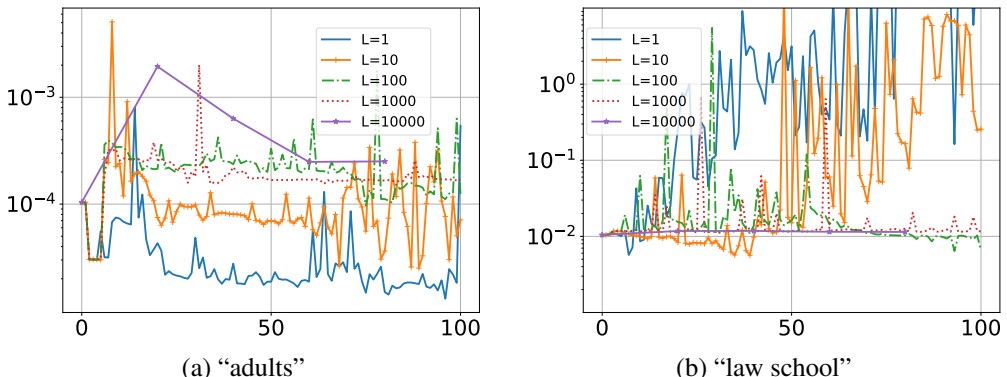

Figure 4: We demonstrate iteration numbers vs. $\|\mathbf{g}(\mathbf{z})\|_2$ for Broyden methods of Lin et al. [23] for adversarial debiasing model on datasets "adults" ($d = 123$, $n = 32561$) and "law school" ($d = 380$, $n = 20427$) with $\mathbf{G}_0 = L\mathbf{I}$. We choose $L$ from $\{1, 10, 100, 1000, 10000\}$

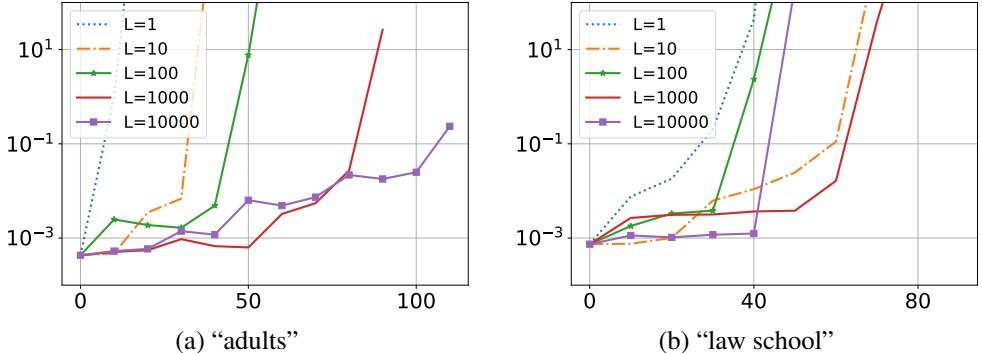

Figure 5: We demonstrate iteration numbers vs. $\|\mathbf{g}(\mathbf{z})\|_2$ for Broyden methods of Ye et al. [46] for adversarial debiasing model on datasets "adults" ($d = 123$, $n = 32561$) and "law school" ($d = 380$, $n = 20427$) with $\mathbf{G}_0 = L\mathbf{I}$. We choose $L$ from $\{1, 10, 100, 1000, 10000\}$