# OpenReview forum: "Quasi-Newton Methods for Saddle Point Problems"
_NeurIPS.cc/2022/Conference — NeurIPS 2022 Accept_

### Official Review · Reviewer_SLU5 · 2022-07-08

**Rating:** 6
**Confidence:** 3
**Soundness:** 3 good
**Presentation:** 3 good
**Contribution:** 3 good

**Summary:**

This paper adapts greedy quasi-Newton method to saddle point problems. The proposed algorithms are based on estimating the square of indefinite Hessian matrix and turn out to have local superlinear convergence.

**Questions:**

Since in the paper authors mentioned two stages convergence behavior, I hope this characteristic can be discussed in the section: experiments.

**Limitations:**

First, this paper requires sufficient smoothness of a strongly-convex and strongly concave function $f$ and proposed algorithm requires the evaluation of squared Hessian, which results in computational complexity. Second, the rate of local super linear convergence is highly related to dimension $n$ which can be very large in practice. If $n$ is large, then the rate will be close to the one for local linear convergence.

**Strengths And Weaknesses:**

The paper is well structured. It is stated clearly for the proposed algorithm that the key is the estimation of squared Hessian. I would like to see a more intensive discussion on how to store the approximate Hessian efficiently. Nevertheless, if all ingredients fit into the memory and are computable, the proposed algorithm is quite efficient, in theory and practice. From all experiments shown in this paper, the proposed algorithms work pretty well eventually. However, in section 5.2, it is better to address more clearly that the objective function satisfies all assumptions.

---

> ### Author Response · Authors · 2022-07-30
> **Response to Reviewer SLU5**
>
> We thank the reviewer for the positive and detailed review as well as the the suggestions for improvement. We use the number of references of the full version in supplementary materials in the following response.
>
> ### Storage
> Our methods require to store the estimator for the square of Hessian matrix, which requires ${\mathcal O}(n^2)$ complexity. The space cost of our methods are the same with quasi-Newton methods for minimization problems [4, 5, 6, 7, 24, 37]. Hence, we think it is reasonable.
> If we expected to reduce the cost of storage, a popular way is using
> limited memory quasi-Newton methods. Section 7 of [29] detailed introduces such type of algorithms for minimization problem.
> To the best of our knowledge, solving minimax problem by limited memory quasi-Newton methods has not been well-studied. We are interested in studying this topic in future work.
>
>
> ### Objective Function in Section 4.2
> Since the convergence results in Section 3.3 are local, we only need to show that $f$ satisfies Assumption 2.1 and 2.2 in a local region around ${\bf z}^* $. Hence, we only needs to consider $({\bf x}, y)$ such that $\|{\bf x}\|\leq D\_1$ and $\|y\|\leq D\_2$ for some $D\_1>0$ and $D_2\>0$.
>
>
> Using the notation in Appendix E.2, we have
> $$
>  \left(2\lambda-\left\|\left\|\frac{\beta y^2}{4m}\sum\_{i=1}^{m}{\bf a}\_i^{\top}{\bf a}\_i\right\|\right\|\right){\bf I}\preceq\nabla^2\_{{\bf x}{\bf x}}f({\bf z})\preceq\left(\|\|\frac{1}{4m}\sum\_{i=1}^{m}{\bf a}\_i^{\top}{\bf a}\_i\|\|+2\lambda\right){\bf I},
> $$
> $$
> 2\gamma\leq -\nabla^2\_{yy} f({\bf z})\leq\left(\frac{1}{4m}\sum\_{i=1}^{m}(\beta{\bf a}\_i^{\top}{\bf x})^2+2\gamma\right).
> $$
> Additionally, the feature ${\bf a}\_i$ is sparse (please refer to Appendix E.2), which means the upper bound of $\left\|\left\|\sum\_{i=1}^{m}{\bf a}\_i^{\top}{\bf a}\_i\right\|\right\|$ is small.
> We can  guarantee $2\lambda-\left\|\left\|\frac{\beta D\_2\^2}{4m}\sum\_{i=1}^{m}{\bf a}\_i\^{\top}{\bf a}\_i\right\|\right\| > 0$ by choosing $\lambda$ properly.
> Theoretically, setting $\mu=\min\left\\{2\lambda-\left\|\left\|\frac{\beta D_2^2}{4m}\sum_{i=1}^{m}{\bf a}_i^{\top}{\bf a}_i\right\|\right\|,2\gamma\right\\}$ can guarantee $f$ satisfies Assumption 2.2.
>
>
>
> The smoothness of $f$ means its Hessian and third-order derivative are continuous. Since we focus on a local closed region, both the norm of Hessian and the third-order derivative are bounded, which guarantees that the gradient and Hessian are Lipschitz-continuous.
>
>
> ### Two Stages Convergence Behavior
> For most of cases, our algorithms enter the local region of superlinear convergence quickly, so that the two-period convergence is not very clear on the figure. Let's have a look at RaBFGSv2-Q in (b) of Figure 1 (the purple curve).
> We can observe two-period convergence behavior clearly.
> The first period (linear convergence) roughly corresponds to the first 2000 rounds of iterations on the figure and the second period (superlinear convergence) roughly corresponds to the later iterations.
> The convergence behavior of first period for RaBFGSv2-Q looks worse than the first-order method (EG), which is reasonable since the linear convergence rate of RaBFGSv2-Q in the first period depends on $\kappa^2$ while the linear convergence rate of EG depends on $\kappa$.
>
>
>
> ### Response to Limitations
>
> 1. The sufficient smoothness is a standard assumption for second-order optimization.
>
> 2. Note that all of proposed algorithms do NOT require evaluate squared Hessian explicitly. All we need in the implementation is computing the Hessian-vector. Hence, it is computationally efficient. In Appendix E, we provide detailed expressions for Hessian-vector operation used in our experiments.
>
>
> 3. We do not think the dependence on $n$ of the superlinear convergence rate is a limitation of our work. Actually, even for minimization problems, the local convergence of quasi Newton methods cannot avoid such dependency on $n$. For example, [37] provides local superlinear convergence rate ${\mathcal O}\left(\left(1-\frac{1}{n\kappa}\right)^{k(k-1)/2}\right)$ for minimization problem.
> Later, [24] shows the local convergence rates of SR1 and BFGS methods can be improved to ${\mathcal O}\left(\left(1-\frac{1}{n}\right)^{k(k-1)/2}\right)$.
> Since minimax problem is more difficult than minimization problem, we think the dependency on $n$ for our algorithms are reasonable.

---

### Official Review · Reviewer_fZXn · 2022-07-09

**Rating:** 6
**Confidence:** 3
**Soundness:** 3 good
**Presentation:** 1 poor
**Contribution:** 3 good

**Summary:**

This paper proposes a randomized Broyden family updates for strongly-convex-strongly concave saddle point problems and show that the algorithm achieves a local superlinear convergence rate. The key idea is to update the square of the Hessian instead of the Hessian itself. Numerical experiments show that the proposed algorithm outperforms classical first-order methods.

**Questions:**

Although the authors do discuss relevant work in the introduction, a related work section is missing from this paper, which makes it unclear which parts of this paper are novel and which parts are mostly based on previous results. Can the authors clarify the key technical difficulties in this paper apart from the need to deal with an indefinite Hessian?

**Strengths And Weaknesses:**

- The authors circumvent the issue of dealing with an indefinite Hessian, which is a nontrival obstacle for saddle point problems, by applying BFGS type updates to the square of the Hessian instead. I am not aware of previous work that have established similar guarantees for quasi-newton methods on saddle point problems, so the analysis presented in this paper can be of interest to the NeurIPS audience.

- My main concern about this paper is the presentation.  In this current form the notations are very dense and often introduced without much motivation. Section 3.1 is particularly difficult to parse as the authors introduces a number of lemmas in previous papers without clearly laying out a purpose for these lemmas. I think the presentation would be greatly improved if the main results are introduced first, and then the necessary lemmas for proving the main results are presented with some motivation.

---

> ### Author Response · Authors · 2022-07-30
> **Response to Reviewer fZXn**
>
> We thank the reviewer for the positive and detailed comments as the the suggestions for improvement. We use the number of references of the full version in supplementary materials in the following response.
>
> ### Presentation
> We thank the reviewer's suggestion for improving the presentation.
> In revision, we first introduced our main result in the first paragraph of Section 3 (line 80-81). See the next paragraph for the presentation of Section 3.1.
>
> ### Connection to Related Work
> In revision, we emphasize that the lemmas in Section 3.1 (line 100) come from the existing analysis of quasi-Newton methods for convex optimization [24, 29, 37], which provides some properties for positive-definite matrix.
> The theoretical results in Section 3.2 and Section 3.3 are novel.
> Most our analysis are different from existing work, since we use the different measure to characterize the convergence (see "Key Technical Difficulties" below). The recursive relationship between ${\bf G}\_k$ and ${\bf G}\_{k+1}$ (such as line 498-500 in Appendix C.3) are mostly based on previous results (Lemma 3.4 in Section 3.1).
>
> ### Key Technical Difficulties
> There are two main technical difficulties. The first is our quasi-Newton methods for minimax problem need to deal with the indefinite Hessian matrix, while the existing quasi-Newton methods for convex optimization only consider the estimation for positive-definite Hessian.
>
> The second is we need to find a proper measure for the convergence analysis.
> Concretely, our algorithm are based on the following update (formula (3) in Section 3):
> $
>     {\bf z}\_{+} = {\bf z}-{\bf G}^{-1}{\hat{\bf H}}({\bf z}){\bf g}({\bf z}).
> $
> Compared with the update of quasi-Newton methods for convex optimization [24, 37], our update rule has an additional term ${\hat{\bf H}}({\bf z})$ between ${\bf G}^{-1}$ and ${\bf g}({\bf z})$, which makes the analysis based on the measure ${\langle {\hat{\bf H}}({\bf z}){\bf g}({\bf z}), {\bf g}({\bf z}) \rangle}^{1/2}$ in [24, 37] is not applicable for our setting. To deal with this issue, our analysis use the different measure $\|\|{\bf g}({\bf z})\|\|$ for convergence analysis.
>
> For other technique contribution, we establish a series of properties for the square of Hessian matrix in Lemma 3.1, 3.10, 3.11 and B.2, which are essential for the convergence analysis of solving saddle point problem by second-order methods.

---

> > ### Comment · Reviewer_fZXn · 2022-08-07
> > **Response to authors**
> >
> > Thank you for the clarifications. I have no further concerns and would like to keep my current score.

---

### Official Review · Reviewer_WrbW · 2022-07-10

**Rating:** 6
**Confidence:** 5
**Soundness:** 3 good
**Presentation:** 3 good
**Contribution:** 2 fair

**Summary:**

In this paper, the authors focus on applying quasi-Newton methods to solve the strongly-convex-strongly-concave saddle point problems. They achieved the explicit local superlinear convergence rates of quasi-Newton methods of Broyden updates. They utilized the technique of estimating the square of the indefinite Hessian matrix instead of the Hessian matrix itself. They also reach the non-asymptotic convergence rate of BFGS method and SR1 method. The authors conduct several numerical experiments on different datasets and these empirical results are consistent with the theoretical results.

**Questions:**

The authors didn't mention the specific results for the DFP method. Is it because that the theoretical results of DPF is not as good as BFGS so there is no need to present the part for DFP?

**Limitations:**

The authors have presented the limitations of this paper in the final conclusion section.

**Strengths And Weaknesses:**

This paper has the following strengths in terms of the originality, quality and clarity:

Originality: this paper provides the first explicit local superlinear convergence rates of quasi-Newton algorithms applied to the saddle point problems. This is the main and independent contribution of this paper.

Quality: the authors provides detailed proof in the theoretical section and the consistent empirical results in the numerical experiments section. These empirical results show that the theoretical results are correct in practice.

Clarity: the paper is organized with clear and clean structure and the language is well-organized so that the readers can easily understand the content of this submission.

However, despite the above advantages, the paper has the one weakness in term of the significance:

This submission presented incremental results of the non-asymptotic superlinear convergence analysis of quasi-Newton methods applied to different problems with different backgrounds. All these proof ideas and techniques are based on the previous work and reference as the authors mentioned in the paper. The authors extend the current superlinear convergence results of quasi-Newton methods applied to the general convex optimization problems to the strongly-convex-strongly-concave saddle point problems. Although the theoretical proof is correct and the authors presented the consistent empirical results, this does not change the essence that this work is an incremental analysis based on the previous similar work. Hence in summary, although this submission is with high quality which improves our understanding of explicit superlinear convergence rates of quasi-Newton methods, it is not totally original and 100% independent. This is a submission with good quality but neither very impressive nor very significant enough.

Some other minor weakness includes:

1. The authors should use t as the time index instead of the k. Because k is very similar to the condition number notation $\kappa$. This can lead to misunderstanding.

2. The authors should use d as the notation of the dimension instead of the n. In the numerical experiment section the authors used d. n is mostly used as the number of functions in the main objective function.

3. The authors should use some other upper-class constant such as $K$ or $C$ to denote the Hessian Lipchitz parameter instead of $L_2$. This notation is much more clear and consistent.

---

> ### Author Response · Authors · 2022-07-30
> **Response to Reviewer WrbW**
>
> We thank the reviewer for the positive and detailed review as well as the the suggestions for improvement. We use the number of references of the full version in supplementary materials in the following response.
>
> ### DFP Method
> The DFP method can be regarded as a special case of the Broyden family methods (Algorithms 2 and 5) with $\tau\_k = 1$.
> Hence, the theoretical results of Broyden family methods in Theorem 3.9 and Theorem 3.17 also holds for the DFP method.
> It is interesting that whether the DFP method has the faster condition-number-free superlinear convergence rate like SR1 and BFGS.
> We believe this is an interesting future direction.
>
> ### Significance
>
> Considering that the Hessian is indefinite in saddle point problem, our algorithms are based on the following update (formula (3) in Section 3): ${\bf z}\_{+} = {\bf z}-{\bf G}^{-1}{\hat{\bf H}}({\bf z}){\bf g}({\bf z}).$
>
> Compared with the update of quasi-Newton methods for minimization problem [24, 37], our update rule has an additional term ${\hat{\bf H}}({\bf z})$ between ${\bf G}^{-1}$ and ${\bf g}({\bf z})$, which makes the analysis based on the measure ${\langle {\hat{\bf H}}({\bf z}){\bf g}({\bf z}), {\bf g}({\bf z}) \rangle}^{1/2}$ in [24, 37] is not applicable for our setting. To deal with this issue, our analysis use the different measure $\|\|{\bf g}({\bf z})\|\|$ for convergence analysis.
> All of above ideas are novel and have not been explored in quasi-Newton methods for minimization problem.
>
>
>
> ### Minus Weakness
> We thank the reviewer's suggestions for notations.  To avoid confusion in the phrase of rebuttal, we have not changed the notations in revision at this stage.  We will improve our notations later.

---

> > ### Comment · Reviewer_WrbW · 2022-08-07
> > **Response to authors**
> >
> > I appreciated the authors for their detailed response and clarification on each comments. I remained my point of view. Thanks.

---

### Official Review · Reviewer_MK9J · 2022-07-11

**Rating:** 8
**Confidence:** 4
**Soundness:** 3 good
**Presentation:** 4 excellent
**Contribution:** 4 excellent

**Summary:**

The authors propose a modifications of well-known Quasi-Newton methods for solving strongly-convex-strongly-concave saddle-points problems with Lipschitz gradietns and Hessians. The key algorithmic novelty is in changing the matrix that is inverted. Unlike to Quasi-Newton methods for minimization problems, where the methods approximate the inverse of the Hessian, the proposed methods estimate the squared Hessian. This modification is important for the analysis of the methods.

Following the analysis in the case of minimization [37, 38, 39, 24], the authors derive two-period local convergence results (one period with linear convergence and one period with superlinear convergence) for general random Broyden family with better rates for BFGS and SR1. These rates are similar to the ones known for minimization up to the replacement of $\kappa$ (condition number) by $\kappa^2$. The worsening in terms of $\kappa$ is expected, since the methods use the inverse of the approximated square of the Hessian.

**Questions:**

1. How do the initial conditions from Theorem 3.17 relate to the ones known for the same methods in minimization? How do they relate to the ones from the concurent works [23] and [46]?
2. Line 582: object --> objective
3. Line 587: continuous --> continuity; object --> objective; "that" should be removed
4. Line 626: should be $\frac{\beta}{M} \leq \frac{1}{4L}$. In fact, this is sufficient since in (56) factor $\mu$ in front of $\frac{\beta}{2M}\rho_k^2$ is missing.
5. Inequality (63): how the last inequality is obtained?



**Limitations:**

All assumptions are properly formulated. There are no potential negative social impact, since the work is purely theoretical.

**Strengths And Weaknesses:**

## Strengths
1. **Novelty and significance.** The results are novel: there are only 2 papers that also propose Quasi-Newton methods for min-max problems (for nonlinear equations), but they consider a different setup and use different ideas. The topic is relevant to ML community. The obtained results are significant.
2. **Clarity.** The results are presented in the clear way, the proofs are also detailed enough and easy to follow.

## Weaknesses
1. **Bad dependence on the condition number.** The derived results suffer from the squared dependence on the condition number $\kappa$. Although I find the proposed approach of approximating the squared Hessian matrix interesting, it would be interesting to see some discussion about the possibility of avoiding such dependence.
2. **Comparison with [46] and [23] is incomplete.** Although the authors discuss the results and assumptions from the concurent works [23, 46], the explicit comparison of rates is missing. I believe such a comparison of rates should be presented in the text. Moreover, in the experiments, it would be interesting to compare with the method from [23] as well: although in [46] it is shown that the method from [23] is inferior to the one from [46], it is shown on particular problems different from the one considered in this paper. Therefore, for completeness, it is important to compare the proposed methods with the one from [23] as well.

---

> ### Author Response · Authors · 2022-07-30
> **Response to Reviewer MK9**
>
> We thank the reviewer for the positive and detailed review as well as the suggestions for improvement.
> We have improved the paper in revision by following reviewers' comments in revision (including the full version in supplementary materials).
> ### Dependence on Condition Number
> We think our results are reasonable, since SCSC minimax problem is more difficult than minimizing strongly-convex function.
> On the other hand, the relationship between our quasi-Newton methods and existing first-order method is similar to the one for minimization problem.
> For our BFGS and SR1 methods, the dependency on $\kappa^2$ only appears on the first period of linear convergence, which matches the convergence rate of gradient descent ascent.
> As an analogy, minimizing strongly-convex function by quasi-Newton methods [24, 37] has $\kappa$ dependency in the first period of linear convergence, which matches the convergence rate of gradient descent.
>
> To improve the dependency on condition number, it is possible to introduce acceleration or extrapolation step for quasi-Newton methods, which may lead to the convergence rate in first period matches the optimal first-order methods, that is $\kappa$ dependency for SCSC minimax and $\sqrt{\kappa}$ dependency for minimizing strongly-convex function.
> We believe this is an interesting topic in future work.
>
> ### Comparison with Concurrent Work [23, 46]
>
> For the convergence rate, we compare our results with [23, 46] in the following table (The measure of [23, 46] is the Euclidean distance $\|\|{\bf z}\_{k}-{\bf z}^* \|\|$. We denote $c=\big\|\big\|\big(\hat{\bf H}^* \big)^{-1}\big\|\big\|$ and $\sigma\_0=\left\|\left\|\big(\hat{\bf H}^* \big)^{-1}(\hat{\bf G}\_0-{\hat{\bf H}}^*)\right\|\right\|$. The measure of ours is the gradient norm $\|\|{\bf g}({\bf z}\_{k})\|\|$. )
>
> |Algorithms    |convergence bound|
> |---------|-------------|
> |[23, Theorem 4.4]|$\left(\frac{7(\sigma_0+\sqrt{cL_2\|{\bf z}_0-{\bf z}^*\|})}{\sqrt{k+k_0}}\right)^{k+k_0}$|
> |[46, Theorem 4.5]| $\left(\frac{4n^2e}{\delta}\right)^{k+k_0}\left(1-\frac{1}{n+1}\right)^{(k+k_0)(k+k_0-1)/4}$|
> |BFGS/SR1 (Alg. 6 and 7)| $\left(1-\frac{1}{n+1}\right)^{k(k-1)/2}\left(\frac{1}{2}\right)^k\left(1-\frac{1}{4\kappa^2}\right)^{k_0}$|
>
> The superlinear convergence rate of [23] is ${\mathcal O}((1/\sqrt{k})^{k})$,  which is worse than ours. The one of [46] is ${\mathcal O}((1-1/(n+1))^{k(k-1)/4})$, which is comparable to ours.
>
>
>
> For the initial condition, we compare our results with [23, 46] and quasi-Newton methods for minimization problem [24, 37] in the table below.
> |Algorithms    |Initial Condition|
> |------------|-------------|
> |[23, Theorem 4.4]|$48L\_2\|\|(\hat{\bf H}^* )^{-1}\|\|\|\|{\bf z}\_0-{\bf z}^* \|\|+\|\|({\hat{\bf H}}^* )^{-1}(\hat{{\bf G}}\_0-{\hat{\bf H}}^ * )\|\|\_{F}\leq \frac{1}{3}$|
> |[46, Theorem 4.5]| $48L_2\|\|({\hat{\bf H}}^ * )^{-1}\|\|\|\|{\bf z}_0-{\bf z}^* \|\|+\|\|({\hat{\bf H}}^{*} )^{-1}(\hat{{\bf G}}_0-{\hat{\bf H}}({\bf z}_0)\|\|\leq \frac{1}{3}$|
> | Ours, Corollary 3.18|  $\|\|\nabla f({\bf z}_0)\|\|\leq \frac{\ln2\mu^5}{16L^3L_2}$|
> |[37, Theorem 4.7] (min)|$\langle{\nabla f({\bf z}_0)},{\nabla^2 f({\bf z}_0)^{-1}\nabla f({\bf z}_0)}\rangle ^{1/2}\leq \frac{\ln (3/2)\mu^{5/2}}{4L_2L}$|
> |[24, Corollary 21] (min)|$\langle{\nabla f({\bf z}_0)},{\nabla^2 f({\bf z}_0)^{-1}\nabla f({\bf z}_0)}\rangle ^{1/2}\leq \frac{\ln (3/2)\mu^{5/2}}{4L_2L}$|
>
>
> Our algorithms only require ${\bf z}_0$ be sufficiently close to the solution, while [23] and [46] need stronger initial condition.
> Concretely, [23] requires $\hat{{\bf G}}_0$ be sufficiently close to ${\hat{\bf H}}^*$  and [46] requires $\hat{{\bf G}}_0$ be sufficiently close to ${\hat{\bf H}}({\bf z}_0)$.
> Finding such $\hat{{\bf G}}_0$ is not easy in general case even if ${\bf z}_0$ is very close to solution.
> In practice, we usually choose $\hat{\bf G}_0 = L{\bf I}$ for their methods.
> The empirical result in Appendix E shows [46] fail to converge by using such $\hat{{\bf G}}_0$ on our model.
> Following the reviewer's suggestion, we also append the experiments to show [23] also fails to converge in Figure 4 of the full version paper in revised supplementary materials.
>
> ### For the Section of Questions
> * 1:  See above.
> * 2 and 3: We'd like to give thanks to the reviewer for pointing out the type error and will fix them in the revision.
> * 4: We thank the reviewer point out this mistake. As the reviewer mentioned, it will not impact the following proof. We will  rewrite (56) as follows
> $$
>         \rho_{k+1}\leq \sigma_k\rho_k +\frac{\beta\mu}{2M} \rho_k^2 \leq \sigma_k\rho_k +\frac{\mu}{8L}\rho_k^2 \leq \sigma_k\rho_k+\frac{1}{8}\rho_k^2.
> $$
> * 5: Recall the definition of $\theta_k$ is $\theta_k\= \sigma_k+2\rho_k$. The last inequality of (63) comes from
> $$
>         2\left(1-\frac{1}{n}\right)\exp (2\rho_k)\geq 1,
> $$
>     where we use the fact $\rho_k \geq 0$ and assume $n \geq 2$.

---

> > ### Comment · Reviewer_MK9J · 2022-08-05
> > **Thank you for the clarifications!**
> >
> > I thank the authors for their detailed response. In particular, I like the explanation why $\varkappa^2$ appears in the complexity bounds (and encourage the authors to add this in the text, since this question is quite natural and appears during the careful read of the paper). The comparison with [23, 46] is now more detailed as it should be in the case of concurent works.
> >
> > Taking the authors response and applied modifications into account I believe that the paper should be accepted to NeurIPS 2022. Moreover, in my opinion, the contribution is significant, solid, with interesting idea and no technical flaws. That is why I have increased my score to 8.

---

### Meta-Review · Area_Chair_kLp2 · 2022-08-23

**Recommendation:** Accept
**Confidence:** Certain

**Metareview:**

Thank you for your submission to NeurIPS. The reviewers unanimously found the work to address an important, relevant problem, and the paper to be clear and generally well-written. All four reviewers unanimously recommend accepting the paper.

The work has obvious impact for the ML community: the idea of rewriting the Newton update $z_+ = z - H^{-1}g$ in terms of the positive definite squared Hessian $z_+ = z - H^{-2} (H g)$ and then using a quasi-Newton scheme to approximate $H^{-2}$ is immediately intuitive and applicable to a wide range of practical problems. The paper provides a rigorous guarantee that such a quasi-Newton scheme converges superlinearly within a neighborhood of the saddle point.

Please incorporate reviewer feedback in preparing the camera ready version. In particular, please take care to include the Comparison to Concurrent Work [23, 46] in your response to reviewer MK9.

**Award:**

No

---

### Decision · Program_Chairs · 2022-09-14

Accept